# Small Generalizable Prompt Predictive Models Can Steer Efficient RL Post-Training of Large Reasoning Models

**Yun Qu** [† 1 2]  **Qi Wang** [1]  **Yixiu Mao** [1]  **Heming Zou** [† 1 2]  **Yuhang Jiang** [1]  **Weijie Liu** [2]  **Clive Bai** [2]  **Kai Yang** [2]  **Yangkun Chen** [2]  **Saiyong Yang** [2]  **Xiangyang Ji** [1]

## Abstract

Reinforcement learning enhances the reasoning capabilities of large language models but often involves high computational costs due to rollout-intensive optimization. Online prompt selection presents a plausible solution by prioritizing informative prompts to improve training efficiency. However, current methods either depend on costly, exact evaluations or construct prompt-specific predictive models lacking generalization across prompts. This study introduces Generalizable Predictive Prompt Selection (GPS), which performs Bayesian inference towards prompt difficulty using a lightweight generative model trained on the shared optimization history. Intermediate-difficulty prioritization and history-anchored diversity are incorporated into the batch acquisition principle to select informative prompt batches. The small predictive model also generalizes at test-time for efficient computational allocation. Experiments across varied reasoning benchmarks indicate GPS's substantial improvements in training efficiency, final performance, and test-time efficiency over superior baseline methods. The code is available at https://github.com/thu-rllab/GPS.

## 1. Introduction

Recent advances have witnessed the impressive reasoning abilities of large language models (LLMs) on complex problem-solving, such as mathematics and programming (Shao et al., 2024; Luo et al., 2025a). Behind the

† Work completed during an internship at Tencent. [1]Department of Automation, Tsinghua University, Beijing, China [2]LLM Department, Tencent, Beijing, China. Correspondence to: Qi Wang <cheemswang@mail.tsinghua.edu.cn>, Saiyong Yang <stevesyang@tencent.com>, Xiangyang Ji <xyji@tsinghua.edu.cn>.

*Proceedings of the 43rd International Conference on Machine Learning*, Seoul, South Korea. PMLR 306, 2026. Copyright 2026 by the author(s).

remarkable success is a key post-training technology, reinforcement learning with verifiable rewards (RLVR) (Jaech et al., 2024; Guo et al., 2025), which prompts the LLM to generate long Chain-of-Thoughts (CoTs) (Wei et al., 2022) and performs policy optimization with verified rewards. Despite its success, RLVR is widely known to be expensive in computations and memory usage, as it requires substantial rollouts for LLM policy evaluation and updates (Zheng et al., 2025b; Lin et al., 2025).

**Pressing demands of online prompt selection.** In LLMs training, data curation is an indispensable process, as the prompt chosen to optimize directly influences convergence and efficiency. In particular, the prompts in RLVR contribute unevenly: overly easy or extremely hard prompts tend to yield vanishing gradients (Yu et al., 2025), while intermediate-difficulty prompts are more informative to update (Zeng et al., 2025b; Chen et al., 2025). Consequently, the naive use of random prompt sampling can be ineffective and prolong the learning process. To this end, several researchers have shifted focus to online prompt selection (Yu et al., 2025; Zhang et al., 2025a; Bae et al., 2025), which evaluates prompt difficulty in a larger candidate set via additional rollouts and selects the subset for RLVR. While such a strategy improves performance and reduces the number of training steps, it incurs substantial computational overhead from additional rollouts (Zheng et al., 2025b).

**Opportunities and challenges of prompt difficulty prediction.** Recognizing the high computational cost of prompt evaluation, recent works (Zheng et al., 2025b; Zhang et al., 2025b; Qu et al., 2025b) propose to predict prompt difficulty from accumulated history and employ selective rollouts to improve training efficiency. However, existing approaches to difficulty estimation, e.g., MoPPS (Qu et al., 2025b), rely on a prompt predictive model (PPM) that tracks difficulty beliefs independently for arbitrary prompt, imposing several limitations. Note that prompt-specific modeling isolates samples during belief updates; hence, only frequently optimized prompts enable reliable difficulty estimates. Other prompts either suffer from stale information or receive few belief updates at all, considering that these prompt-specific PPMs cannot generalize across prompts. Moreover, they

neglect the batch-level structure in prompt selection, leading to redundant training signals within selected batches and limited exploration of the prompt space. All of these raise the research question:

*Can we design a lightweight yet generalizable PPM, together with more effective subset acquisition strategies, to achieve efficient optimization?*

**Small generalizable PPMs and comprehensive batch acquisition boost computational efficiency.** This work positively answers the research question and proposes generalizable predictive prompt selection (GPS), which designs a generalizable PPM and crafts a principled batch acquisition strategy in RLVR. (i) Unlike isolated prompt difficulty modeling (Qu et al., 2025b; Zhang et al., 2025b; Zheng et al., 2025b), we construct a lightweight generative PPM that exploits the entire optimization histories and shares experience across prompts. This way derives the generalizable difficulty prior and improves the accuracy of the prompt difficulty estimate through temporal extrapolation. (ii) Armed with the generalizable PPM, we incorporate the intermediate difficulty prioritization and history-anchored diversity principles into the acquisition criteria for batch selection. This design aims to reduce sampling redundancy and preserve prompt coverage. In addition, the proposed generalizable PPM reserves the potential of guiding efficient test-time computation allocation.

**Contributions and Empirical Findings.** The primary contribution of this work is threefold:

1. We develop a generalizable PPM that leverages the shared optimization history to generalize difficulty prediction across prompts at negligible computational cost for RLVR.

2. The proposed prompt batch acquisition strategy combines difficulty guidance with history-aware diversity to reduce redundancy and improve prompt coverage.

3. The generalizable PPM after RL post-training supports the test-time computation allocation, improving performance under compute budget constraints.

Extensive experiments on large-scale mathematical and logical reasoning benchmarks across diverse LLM backbones witness several advantages of our scheme in both training and test time: (i) GPS reliably predicts prompt difficulty and improves the quality of the selected prompt batch. (ii) GPS accelerates RL post-training, delivering up to $2.0\times$ speedup over uniform sampling while consistently improving performance. Moreover, GPS matches or outperforms evaluation-based selection with up to $69\%$ lower rollout cost. (iii) At test time, the learned PPM enables effective computation reallocation, reducing inference cost by up to $36.4\%$ without performance loss or improving accuracy by up to $3.2\%$ under fixed budgets.

## 2. Preliminaries

### 2.1. Notations

We consider reasoning tasks in which prompts $\tau$ take the form of mathematical or logical problems, e.g., "If $991 + 993 + 995 + 997 + 999 = 5000 - N$, then $N$?" from Deepscaler (Luo et al., 2025b). Let $\mathcal{T} = \{\tau_i\}_{i=1}^N$ denote the full prompt pool. At iteration $t$, the LLM $\pi_{\boldsymbol{\theta}_t}$ serves as the policy. In each iteration, a batch of prompts $\mathcal{T}_t^{\mathcal{B}} = \{\tau_{t,i}\}_{i=1}^{\mathcal{B}} \subset \mathcal{T}$ is selected to generate rollouts, i.e., long CoT responses, for optimization.

For each prompt $\tau \in \mathcal{T}_t^{\mathcal{B}}$, the LLM generates $k$ independent responses $\boldsymbol{y}_t^\tau = \{y_{t,j}^\tau\}_{j=1}^k$, with each verified by a reward function $r(\tau, y)$ to yield the reward set $\boldsymbol{r}_t^\tau = \{r_{t,j}^\tau\}_{j=1}^k$. Following common practice (Guo et al., 2025), we focus on binary correctness rewards, while our formulation does not rely on this restriction (Appendix E.5). We denote $\gamma_t^\tau = \mathbb{E}[r_t^\tau]$ as the latent success rate of prompt $\tau$ at step $t$, which acts as an intrinsic difficulty attribute of the prompt and represents the probability that the current model solves $\tau$. The batch-level feedback at $t$-th step is $\mathcal{R}_t^{\mathcal{B}} = \{\boldsymbol{r}_t^{\tau_{t,i}}\}_{i=1}^{\mathcal{B}}$, corresponding to the generative process:

$$p(\mathcal{R}_t^{\mathcal{B}}|\mathcal{T}_t^{\mathcal{B}}, \boldsymbol{\theta}_t) = \int \prod_{i=1}^{\mathcal{B}} p(\boldsymbol{r}_t^{\tau_{t,i}}|\gamma_t^{\tau_{t,i}}) p(\gamma_t^{\tau_{t,i}}|\tau_{t,i}, \boldsymbol{\theta}_t) d\gamma_t^{\tau_{t,i}}.$$

We define the observation history of prompt $\tau$ as $H_t^\tau = \{\boldsymbol{r}_j^\tau | \tau \in \mathcal{T}_j^{\mathcal{B}} \text{ and } j \leq t\}$. The full optimization history is defined as the collection of all prompt-specific history $H_t = \bigcup_{\tau \in \mathcal{T}} H_t^\tau = \{\mathcal{T}_i^{\mathcal{B}}, \mathcal{R}_i^{\mathcal{B}}\}_{i=1}^t$.

For brevity, we use $\tau$ to denote both the natural language prompt and its corresponding embedding representation. The choice of encoding mechanism is orthogonal to our core studies, and we adopt the WordLlama toolkit implementation (Miller, 2024) to ensure efficient processing, with further discussion provided in Appendix B.3.

### 2.2. Reinforcement Learning with Verifiable Rewards

RLVR aims to optimize the LLM policy $\pi_{\boldsymbol{\theta}}$ to maximize the expected verifiable reward:

$$\max_{\boldsymbol{\theta}} \ \mathbb{E}_{\tau \sim \mathcal{T}, \ y \sim \pi_{\boldsymbol{\theta}}(\cdot|\tau)} \big[r(\tau, y)\big], \qquad (1)$$

where $\pi_{\boldsymbol{\theta}}(y|\tau)$ is the model's conditional distribution over responses for prompt $\tau$, and $r(\tau, y)$ is a verifiable reward evaluating the quality of response $y$.

**Group Relative Policy Optimization (GRPO).** Shao et al. (2024) introduces GRPO to stablize RL post-training via the group-normalized advantage while removing the value

network used in PPO (Schulman et al., 2017):

$$
\mathcal{J}_{\text{GRPO}}(\boldsymbol{\theta}) = \mathbb{E}_{\tau \sim \mathcal{T}_t^{\mathcal{B}}, \{y_i^{\tau}\}_{i=1}^{k} \sim \pi_{\boldsymbol{\theta}_{\text{old}}}(\cdot|\tau)} \left[ \frac{1}{k} \sum_{i=1}^{k} \frac{1}{|y_i^{\tau}|} \sum_{t=1}^{|y_i^{\tau}|} (\min \big(
$$

$$
\rho_{i,t} \cdot \hat{A}_i, \ \text{clip}(\rho_{i,t}, 1 - \epsilon, 1 + \epsilon) \cdot \hat{A}_i \big) - \beta \cdot \text{KL}(\pi_{\boldsymbol{\theta}}||\pi_{\text{ref}}) \big) \right]
$$

(2)

where $\rho_{i,t} = \frac{\pi_{\boldsymbol{\theta}}(y_{i,t}|\tau, y_{i,<t})}{\pi_{\boldsymbol{\theta}_{\text{old}}}(y_{i,t}|\tau, y_{i,<t})}$ is the importance ratio, and $\pi_{\text{ref}}$ is a fixed reference policy. The KL divergence term penalizes deviation from $\pi_{\text{ref}}$, with $\beta$ controlling the regularization strength. The group-relative advantage for the $i$-th response is computed as

$$
\hat{A}_i = \frac{r_i^{\tau} - \text{mean}(\{r_i^{\tau}\}_{i=1}^{k})}{\text{std}(\{r_i^{\tau}\}_{i=1}^{k})}.
$$

(3)

### 2.3. Online Prompt Selection for RLVR

RLVR enhances LLM reasoning but comes with high computational costs, driving the need for prompt curation. Previous studies show that prompts affect optimization unevenly. For algorithms like GRPO, rewards with zero variance can lead to vanishing policy gradients (Yu et al., 2025; Zhang et al., 2025b). In contrast, prompts of intermediate difficulty provide more informative training signals (Chen et al., 2025; Bae et al., 2025; Zeng et al., 2025b). To mitigate ineffective prompts and enhance training, online prompt selection methods (Yu et al., 2025; Qu et al., 2025b; Zhang et al., 2025a; Zheng et al., 2025b) have been proposed to adaptively choose prompt batches during optimization. The key to the prompt evaluation is to precisely estimate the success rate $\gamma_t^{\tau_t}$ under an arbitrary $\tau_t$ and $\boldsymbol{\theta}_t$ at the $t$-th iteration.

**Evaluation-based selection.** Methods such as Dynamic Sampling (DS) (Yu et al., 2025) and SPEED-RL (Zhang et al., 2025a) over-sample and evaluate a candidate set $\mathcal{T}_t^{\hat{\mathcal{B}}}$ of size $\hat{\mathcal{B}}$ with real model rollouts, and then select the batch:

$$
\mathcal{T}_t^{\mathcal{B}} = \left\{ \tau \in \mathcal{T}_t^{\hat{\mathcal{B}}} \ \big| \ \text{std}(\{r_i^{\tau}\}_{i=1}^{k}) > 0 \right\}.
$$

(4)

This approach ensures informative prompts but exhausts computations from large-scale exact prompt evaluations (Zheng et al., 2025b).

**Prediction-based selection.** To circumvent extra prompt evaluation, several methods (Zhang et al., 2025b; Zheng et al., 2025b; Gao et al., 2025) adopt the strategy of predicting prompt difficulty. This requires constructing a series of PPMs as $p(\gamma_t^{\tau}|\tau, \boldsymbol{\theta}_t)$ iteratively. Take MoPPS (Qu et al., 2025b) as example, it employs a Beta posterior for each prompt and utilizes specific history data to estimate success rates. Nevertheless, the PPM in MoPPS hardly keeps pace with the evolving model state $\boldsymbol{\theta}_t$ and treats prompts independently. These raise the generalization issue due to the failure of sharing information across prompts and update delay degrades the success rate accuracy.

**Algorithm 1** Generalizable Predictive Prompt Selection (GPS)

**Input:** Prompt pool $\mathcal{T} = \{\tau_i\}_{i=1}^{N}$; Batch size $\mathcal{B}$; LLM $\pi_{\boldsymbol{\theta}_1}$ with parameters $\boldsymbol{\theta}_1$; Generalizable PPM with parameters $(\boldsymbol{\psi}_1, \boldsymbol{\eta}_1, \boldsymbol{\phi}_1)$; Diversity weight $\lambda$; Training steps $T$.

**Output:** LLM $\pi_{\boldsymbol{\theta}}$

Initialize history $H_0 \leftarrow \emptyset$

**for** $t = 1$ **to** $T$ **do**

    // Prompt Difficulty Prediction

    **foreach** $\hat{\tau} \in \mathcal{T}$ **do**

        Estimate $\hat{\gamma}_t^{\hat{\tau}}$ via Eq. (10) using $H_{t-1}$

    // Prompt Batch Selection

    Initialize selected batch $\mathcal{T}_t^{\text{partial}} \leftarrow \emptyset$

    **while** $|\mathcal{T}_t^{\text{partial}}| < \mathcal{B}$ **do**

        Select a prompt $\tau^{\star}$ via Eq. (14)

        $\mathcal{T}_t^{\text{partial}} \leftarrow \mathcal{T}_t^{\text{partial}} \cup \{\tau^{\star}\}$

    $\mathcal{T}_t^{\mathcal{B}} \leftarrow \mathcal{T}_t^{\text{partial}}$

    **foreach** $\tau \in \mathcal{T}_t^{\mathcal{B}}$ **do**

        Generate responses $\boldsymbol{y}_t^{\tau} = \{y_{t,j}^{\tau}\}_{j=1}^{k}$ using $\pi_{\boldsymbol{\theta}_t}$

        Compute rewards $\boldsymbol{r}_t^{\tau} = \{r_{t,j}^{\tau}\}_{j=1}^{k}$

    // Policy Update in RLVR

    Update LLM $\boldsymbol{\theta}_t \rightarrow \boldsymbol{\theta}_{t+1}$ with $\{(\tau, \boldsymbol{y}_t^{\tau}, \boldsymbol{r}_t^{\tau})\}_{\tau \in \mathcal{T}_t^{\mathcal{B}}}$

    // History and PPM Update

    $H_t \leftarrow H_{t-1} \cup \{(\tau, \boldsymbol{r}_t^{\tau})\}_{\tau \in \mathcal{T}_t^{\mathcal{B}}}$

    Update $(\boldsymbol{\psi}_t, \boldsymbol{\eta}_t, \boldsymbol{\phi}_t)$ by maximizing ELBO in Eq. (8)

## 3. Method

This section presents GPS to construct a generalizable PPM, along with a principled acquisition criterion for efficient RL post-training. Meanwhile, we show that test-time compute allocation can benefit from the online learned PPM.

### 3.1. The Curse of Prompt-Specific PPMs

The family of prompt-specific PPMs (Zhang et al., 2025b; Zheng et al., 2025b; Qu et al., 2025b) estimates $\gamma_t^{\tau}$ with prompt-specific histories, maintaining independent posteriors per prompt:

$$
p(\{\gamma_t^{\tau}\}_{\tau \in \mathcal{T}}|H_{t-1}) = \prod_{\tau \in \mathcal{T}} p(\gamma_t^{\tau}|\tau, H_{t-1}^{\tau}).
$$

(5)

While this formulation is computationally efficient, it raises significant concerns when handling large-scale prompts and rapidly changing LLM dynamics:

- **Cross-Prompt Generalization Bottleneck.** Independent modeling in Eq. (5) implicitly assumes that difficulty is non-transferable, precluding information sharing between related prompts. This encounters a cold-start bottleneck. Reliable estimates are only available

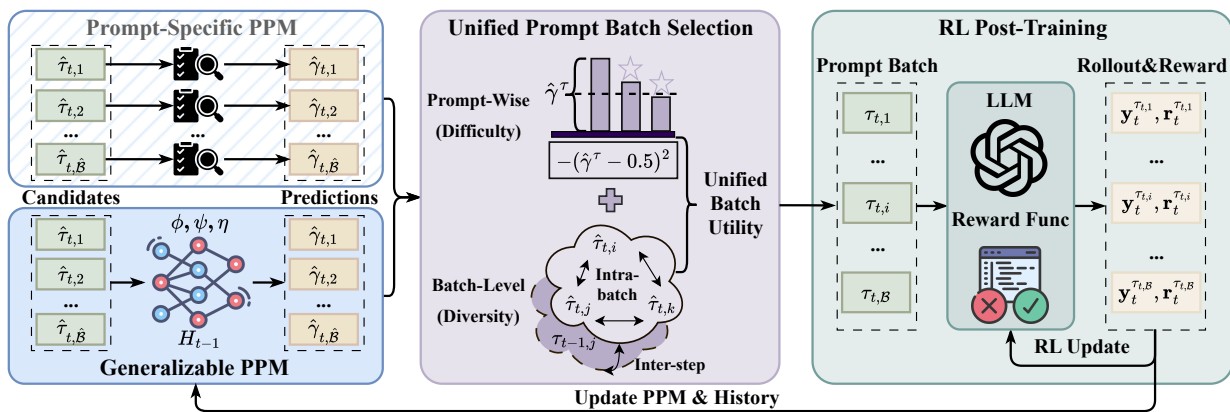

*Figure 1.* Framework overview. Unlike prompt-specific modeling like MoPPS (Qu et al., 2025b), GPS proposes a generalizable prompt predictive model (PPM) to estimate difficulty and track LLM evolution. The comprehensive prompt batch acquisition accounts for intermediate difficulty prioritization and batch-level diversity, improving RL post-training efficiency while mitigating PPM overfitting.

for frequently sampled prompts. In high-dimensional prompt spaces, independent estimators often collapse toward uninformative priors, failing to leverage the semantic structure of the prompt.

- **Accumulated Estimation Error from Delayed Reward Signals.** Standard approaches often assume local stationarity, extrapolating future performance from past difficulty in the absence of sufficient prompt-specific observations. However, because the underlying model evolves continuously during training, prompt difficulty is inherently non-stationary. Without a mechanism to model these dynamics, predictors tend to suffer from lagged adaptation, failing to perceive performance shifts before they occur.

Moreover, such methods also fail to generalize to unseen prompts after training, as summarized in Table 1.

Note that the prompt difficulty depends jointly on the prompt $\tau$ itself and the evolving $\boldsymbol{\theta}_t$, while the accumulated history $H_{t-1}$ reflects the change of LLM dynamics. Hence, distinguished from Eq. (5), this work turns to a more natural way of estimating the prompt difficulty, which uses the complete history $H_{t-1}$ in modeling, i.e.,

$$p(\gamma_t^\tau | \tau, H_{t-1}), \ \forall \tau \in \mathcal{T}. \quad (6)$$

For an arbitrary prompt $\tau$, this approach can circumvent delayed reward signals and potentially improve predictive accuracy with the help of the optimization signals from other relevant prompts.

### 3.2. Generalizable Prompt Predictive Model

Building on the previous analysis, we investigate how to construct generalizable PPMs. The basic idea is to leverage optimization history and compress difficulty-aware information into the latent variable shared across prompts.

**Generative Modeling of PPMs with Latent Variables.** Here, we treat the prompt feedback and associated optimization signals at the $t$-th step as the conditional generative result from previous accumulated experience $H_{t-1}$. Specifically, a global latent variable $\boldsymbol{z}_t$, referred to as the *difficulty context*, is introduced to provide foresight of generalizable prompt difficulty information from $H_{t-1}$. Then the marginal distribution of the whole optimization history can be factorized into:

$$p(H_{0:T}) = p(H_0) \int \prod_{t=1}^{T} p_{\boldsymbol{\psi}}(H_t|\boldsymbol{z}_t) p_{\boldsymbol{\eta}}(\boldsymbol{z}_t|H_{t-1}) d\boldsymbol{z}_{1:T}, \quad (7)$$

where $p_{\boldsymbol{\eta}}(\boldsymbol{z}_t|H_{t-1})$ is a conditional prior, and $p_{\boldsymbol{\psi}}(H_t|\boldsymbol{z}_t) \triangleq \prod_{i=1}^{\mathcal{B}} p_{\boldsymbol{\psi}}(\gamma_t^{\tau_{t,i}}|\tau_{t,i}, \boldsymbol{z}_t)$ is a shared likelihood model for prompt difficulty estimation. The function of global latent variables is akin to classical conditional generative modeling (Sohn et al., 2015; Garnelo et al., 2018; Wang & Van Hoof, 2020; 2022a;b; Wang et al., 2023a) to represent the context that allows information to transfer across prompts, while the evolving prior enables predictions to adapt to the non-stationary optimization trajectory.

**Variational Inference.** To handle the intractability of the marginal likelihood, we introduce a variational posterior $q_{\boldsymbol{\phi}}(\boldsymbol{z}_t|H_t)$. Applying Jensen's inequality yields the evidence lower bound (ELBO):

$$\ln p(H_t|H_{t-1}) = \ln \mathbb{E}_{q_{\boldsymbol{\phi}}(\boldsymbol{z}_t|H_t)} \left[ \frac{p_{\boldsymbol{\eta}}(\boldsymbol{z}_t|H_{t-1}) p_{\boldsymbol{\psi}}(H_t|\boldsymbol{z}_t)}{q_{\boldsymbol{\phi}}(\boldsymbol{z}_t|H_t)} \right]$$
$$\geq \mathbb{E}_{q_{\boldsymbol{\phi}}(\boldsymbol{z}_t|H_t)} \left[ \ln p_{\boldsymbol{\psi}}(H_t|\boldsymbol{z}_t) \right] - D_{\mathrm{KL}} \left[ q_{\boldsymbol{\phi}}(\boldsymbol{z}_t|H_t) \| p_{\boldsymbol{\eta}}(\boldsymbol{z}_t|H_{t-1}) \right]$$
$$= \mathcal{L}_{\mathrm{ELBO}}(\boldsymbol{\psi}, \boldsymbol{\phi}, \boldsymbol{\eta})$$
$$(8)$$

Maximizing the ELBO jointly optimizes: (i) the decoder $p_{\boldsymbol{\psi}}$, which captures the shared mapping from context to difficulty; (ii) the encoder $q_{\boldsymbol{\phi}}$, which extracts current contextual information; and (iii) the history-dependent prior $p_{\boldsymbol{\eta}}$, which

*Table 1.* Comparison of typical online prompt selection methods.

| Method | Prompt Evaluation | Difficulty Modeling | Prompt Batch Selection | Compute Efficiency | Dataset Scalability | Test-time Computation |
|--------|-------------------|---------------------|------------------------|--------------------|--------------------|-----------------------|
| DS | Exact | - | Threshold Filtering | ✗ | ✓ | ✗ |
| MoPPS | Predictive | Independent | Max-Sum (Top-$\mathcal{B}$) | ✓ | ✗ | ✗ |
| GPS (Ours) | Predictive | Global | Max-Sum Diversity | ✓ | ✓ | ✓ |

acts as a temporal summarizer of the optimization history. Implementation details are deferred to Appendix D.4.

**Predictive Distribution.** The difficulty of a candidate prompt $\tau_t$ at step $t$ is quantified via the predictive distribution

$$p(\gamma|\tau_t, H_{t-1}) = \int p_{\psi}(\gamma|\tau_t, \boldsymbol{z}_t) p_{\boldsymbol{\eta}}(\boldsymbol{z}_t|H_{t-1}) d\boldsymbol{z}_t. \quad (9)$$

In practice, we use Monte Carlo approximation:

$$\hat{\gamma}_t^{\tau_t} \approx \frac{1}{M} \sum_{m=1}^{M} p_{\psi}(\gamma|\tau_t, \boldsymbol{z}_t^{(m)}), \text{ with } \boldsymbol{z}_t^{(m)} \sim p_{\boldsymbol{\eta}}(\cdot|H_{t-1}). \quad (10)$$

This mechanism enables the model to infer the difficulty of even unvisited prompts, while the time-varying prior $p_{\boldsymbol{\eta}}$ ensures the predictions remain synchronized with the evolving policy.

Further, we provide a theoretical justification for global difficulty modeling, suggesting that conditioning on full optimization history yields a strictly lower predictive mean squared error than independent estimators based solely on per-prompt observations. A formal proof is provided in Appendix C.

**Theorem 3.1** (Better Prediction with Shared History). *Let $\hat{\gamma}^{\tau,\text{ind}} := \mathbb{E}[\gamma_t^{\tau}|H_{t-1}^{\tau}]$ be the optimal predictor based only on prompt-specific history, and $\hat{\gamma}^{\tau,\text{shr}} := \mathbb{E}[\gamma_t^{\tau}|H_{t-1}]$ be the optimal predictor based on the full optimization history, where $H_{t-1}^{\tau} \subset H_{t-1}$. The prediction risk $\mathcal{R}(\hat{\gamma}) := \mathbb{E}[(\hat{\gamma} - \gamma_t^{\tau})^2]$ admits the orthogonal decomposition*

$$\mathcal{R}(\hat{\gamma}^{\tau,\text{shr}}) = \mathcal{R}(\hat{\gamma}^{\tau,\text{ind}}) - \mathcal{C}(\tau), \quad (11)$$

*where $\mathcal{C}(\tau) = \mathbb{E}\left[\left(\hat{\gamma}^{\tau,\text{shr}} - \hat{\gamma}^{\tau,\text{ind}}\right)^2\right] \geq 0$ is the estimation gap. Moreover, $\mathcal{C}(\tau) > 0$ if and only if $H_{t-1}$ provides non-redundant predictive information about $\gamma_t^{\tau}$ beyond $H_{t-1}^{\tau}$.*

### 3.3. Difficulty-Diversity Unified Prompt Batch Selection

Prior methods (Chen et al., 2025; Qu et al., 2025b; Gao et al., 2025) select prompts based on independent prompt-wise scoring, without considering batch-level structure. Instead, we formulate prompt selection as a batch-level optimization problem that explicitly balances prompt-wise difficulty with batch-level diversity, seeking informative and diverse prompt batches.

**Unified Batch Utility.** Specifically, at training step $t$, our objective is to select a subset $\mathcal{T}_t^{\mathcal{B}} \subset \mathcal{T}$ that maximizes the following batch-level utility:

$$\arg\max_{\mathcal{T}_t^{\mathcal{B}} \subset \mathcal{T}} U(\mathcal{T}_t^{\mathcal{B}}) = \underbrace{\sum_{\tau \in \mathcal{T}_t^{\mathcal{B}}} u(\hat{\gamma}_t^{\tau})}_{\text{Difficulty Utility}} + \lambda \cdot \underbrace{D(\mathcal{T}_t^{\mathcal{B}}; \mathcal{T}_{t-1}^{\mathcal{B}})}_{\text{History-Anchored Diversity}}, \quad (12)$$

where $u(\hat{\gamma}_t^{\tau})$ measures the informativeness of an individual prompt, $D(\cdot)$ promotes diversity at the batch level, and $\lambda > 0$ controls the trade-off. Following common practice (Xu et al., 2025; Sun et al., 2025), we prioritize prompts with intermediate difficulty and adopt $u(\hat{\gamma}_t^{\tau}) = -(\hat{\gamma}_t^{\tau} - 0.5)^2$. This specific choice is highly effective for binary-reward settings, as targeting a $0.5$ success rate theoretically maximizes reward variance for stronger learning signals while naturally inducing an easy-to-hard curriculum. Importantly, GPS is flexible and can seamlessly accommodate alternative criteria; see Appendix E.5 for an example.

The diversity term encourages the batch to span a broad and non-repetitive region of the prompt space:

$$D(\mathcal{T}_t^{\mathcal{B}}; \mathcal{T}_{t-1}^{\mathcal{B}}) = \underbrace{\sum_{\tau_i, \tau_j \in \mathcal{T}_t^{\mathcal{B}}} \text{dist}(\tau_i, \tau_j)}_{\text{Intra-batch Dispersion}} + \underbrace{\sum_{\tau \in \mathcal{T}_t^{\mathcal{B}}} \sum_{\tau' \in \mathcal{T}_{t-1}^{\mathcal{B}}} \text{dist}(\tau, \tau')}_{\text{Inter-step Exploration}}, \quad (13)$$

where $\text{dist}(\cdot, \cdot)$ is a distance measure. This provides dual benefits: it (i) improves LLM training by reducing redundant signals (Jin et al., 2025b; Qu et al., 2025c) and (ii) mitigates overfitting of the PPM by encouraging exposure to a broader region of the prompt space, as discussed in Appendix B.2.

**Greedy Search Prompt Batch.** Maximizing $U(\mathcal{T}_t^{\mathcal{B}})$ is a max-sum diversification problem, which is NP-hard (Borodin et al., 2012). We adopt a standard greedy approximation (Borodin et al., 2012; Wang et al., 2023b), which iteratively selects the prompt with the largest marginal gain

$$\tau^{\star} = \arg\max_{\tau \in \mathcal{T} \setminus \mathcal{T}_t^{\text{partial}}} \left[ U(\mathcal{T}_t^{\text{partial}} \cup \{\tau\}) - U(\mathcal{T}_t^{\text{partial}}) \right], \quad (14)$$

where $\mathcal{T}_t^{\text{partial}}$ denotes the batch constructed so far. This greedy heuristic effectively balances selection quality with computational efficiency, enabling scalable prompt selection for large-scale LLM training.

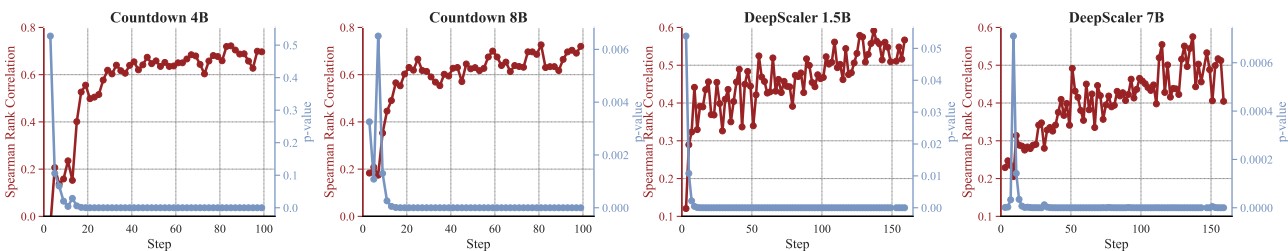

*Figure 2.* Spearman's rank correlation and *p*-value during training between predicted prompt difficulty and empirical success rate.

### 3.4. Extension to Test-Time Computation Allocation

Recent work shows that uniformly allocating test-time computation leads to inefficient budget usage (Damani et al., 2024; Wang et al., 2025c). Considering *Best-of-N* sampling (Cobbe et al., 2021; Lightman et al., 2023), the marginal gain of one extra sample to prompt $\tau$ is

$$\Delta_k(\tau) = \text{pass@(k + 1)}(\tau) - \text{pass@k}(\tau) = (1-\gamma^\tau)^k\gamma^\tau, \quad (15)$$

which motivates difficulty-aware allocation. Fortunately, unlike prompt-specific modeling, generalizable PPM captures a global and transferable notion of prompt difficulty, making it feasible to generalize across unseen prompts for guiding test-time computation allocation.

Specifically, following the allocation strategy of (Damani et al., 2024), we partition test prompts into three bins based on predicted difficulty and allocate lower budgets to easy or likely-unsolvable prompts, while assigning more computation to promising prompts that are challenging yet solvable (details in Appendix D.5). This reuse does not introduce additional training and can offer improvements over uniform allocation under a fixed budget, naturally complementing the training-time framework.

## 4. Experiments

### 4.1. Experimental Setup

We evaluate GPS on representative reasoning tasks: **mathematical** and **logical reasoning**, using large-scale benchmarks. To assess the generality, we experiment with a **diverse set of LLM backbones** spanning different model sizes, including both base models and distilled models. For RLVR algorithm, we primarily adopt the GRPO algorithm within the verl framework (Sheng et al., 2024), though GPS is compatible with other algorithms as shown in Sec. 4.3.1. Model performance is measured by test accuracy, reported as the average pass@1 over multiple rollouts per problem, where the number of generations varies across benchmarks.

**Mathematical Reasoning.** We train on the Deep-Scaler dataset (Luo et al., 2025b), which contains 40.3k competition-level math problems. Following prior work (Luo et al., 2025b; Qu et al., 2025b), we use DeepSeek-

R1 distilled models: R1-Distill-1.5B (DSR-1.5B) and R1-Distill-7B (DSR-7B). Evaluation is conducted on a suite of standard mathematical benchmarks, namely AIME24, AMC23, MATH500 (Lightman et al., 2023), Minerva Math (Minerva.) (Lewkowycz et al., 2022), and OlympiadBench (Olymp.) (He et al., 2024). To assess out-of-distribution generalization, we evaluate on general reasoning benchmarks: MMLU-Pro (Wang et al., 2024b), ARC-c (Clark et al., 2018), and GPQA-diamond (GPQA) (Rein et al., 2024).

**Logical Reasoning.** We adopt the Countdown Number Game, a combinatorial arithmetic reasoning task that requires composing given numbers with basic operations to match a target value. Training is performed on a 20k-instance subset of the Countdown-34 (CD34) dataset (Pan et al., 2025). Models are evaluated on both CD34 and a more challenging variant, Countdown-4 (CD4). We use two base LLMs, Qwen3-4B-Base and Qwen3-8B-Base (Yang et al., 2025), and one instruct LLM Llama-3.2-3B-Instruct (Grattafiori et al., 2024) in Appendix E.7.

**Baselines.** We compare GPS with several representative sampling strategies: (1) **Uniform** samples prompts uniformly; (2) **MoPPS** (Qu et al., 2025b) maintains prompt-specific Beta posteriors to estimate difficulty from historical observations; (3) **PCL** (Gao et al., 2025) estimates prompt difficulty using a LLM; (4) **GRESO** (Zheng et al., 2025b) maintains per-prompt historical reward statistics to guide probabilistic filtering; and (5) **Dynamic Sampling (DS)** (Yu et al., 2025), which oversamples prompts and filters out uninformative ones using exact evaluation. We consider **DS as an Oracle** baseline since it relies on real evaluation. Our focus is on reducing computational overhead relative to DS rather than outperforming it in accuracy.

Additional implementation details are provided in Appendix D, together with additional experimental settings and extended evaluations in Appendix E and prompt examples in Appendix F.

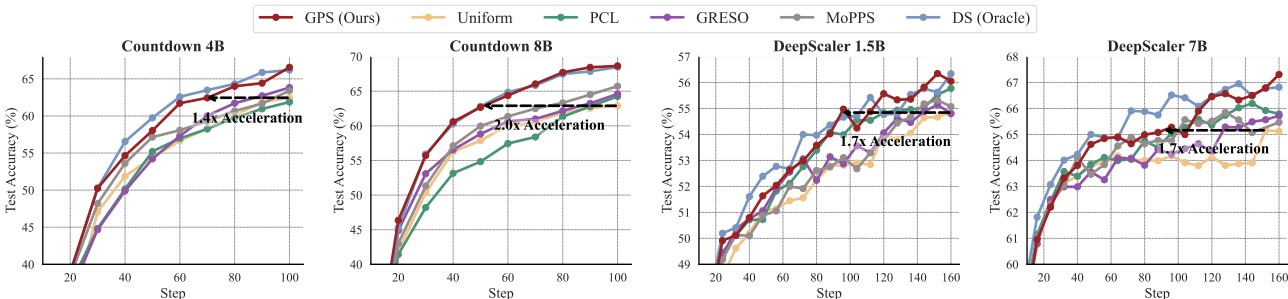

*Figure 3.* Training curves of GPS and baselines across different scenarios and backbone models versus training steps. DS serves an oracle baseline with respect to training steps, but incurs substantially higher rollout costs. Training curves plotted against the number of rollouts are provided in Fig. 7.

*Table 2.* Evaluation on mathematics benchmarks. '+' denotes finetuning with the corresponding method. 'Avg.' reports the average accuracy, and 'Runtime' indicates total training time. **Bold** and underlined values indicate the best and second-best results, respectively.

| Method | In-Distribution | | | | | | Out-of-Distribution | | | | |
| --- | --- | --- | --- | --- | --- | --- | --- | --- | --- | --- | --- |
| | **MATH500** Avg@1 | **Olympiad.** Avg@1 | **Minerva.** Avg@4 | **AMC23** Avg@32 | **AIME24** Avg@32 | **Avg.↑** | **MMLU-Pro** Avg@8 | **ARC-c** Avg@8 | **GPQA** Avg@8 | **Avg.↑** | **Runtime↓** |
| **DSR-1.5B** | 77.8 | 36.6 | 25.6 | 57.7 | 20.8 | 43.7 | 21.2 | 43.1 | 22.8 | 29.0 | - |
| +Uniform | 86.2 | 49.0 | 30.8 | 76.1 | 32.4 | 54.9 | 22.3 | 45.4 | 27.5 | 31.7 | 16h |
| +PCL | 84.8 | 49.6 | 33.0 | 77.7 | 34.2 | 55.8 | 24.0 | 46.3 | 28.5 | 33.0 | 17h |
| +GRESO | 86.2 | 50.6 | 30.7 | 77.1 | 31.5 | 55.2 | 24.7 | 46.7 | 26.4 | 32.6 | 27h |
| +MoPPS | 86.2 | 49.0 | 29.9 | 77.8 | 34.0 | 55.4 | 21.4 | 44.6 | 27.5 | 31.2 | 17h |
| +DS (Oracle) | 87.2 | 51.0 | 30.4 | 79.2 | 34.7 | **56.5** | 24.5 | 46.7 | 26.8 | 32.7 | 30h |
| +GPS (Ours) | 88.0 | 51.3 | 31.3 | 78.1 | 33.8 | **56.5** | 23.3 | 46.9 | 30.4 | **33.5** | 16h |
| **DSR-7B** | 87.2 | 47.5 | 35.8 | 76.5 | 38.8 | 57.1 | 50.5 | 76.1 | 19.2 | 48.6 | - |
| +Uniform | 91.6 | 57.9 | 37.5 | 89.5 | 50.9 | 65.5 | 49.8 | 74.1 | 17.3 | 47.1 | 40h |
| +PCL | 93.2 | 58.0 | 38.4 | 90.1 | 52.8 | 66.5 | 50.6 | 74.4 | 22.4 | 49.1 | 50h |
| +GRESO | 91.8 | 58.2 | 39.3 | 89.7 | 49.8 | 65.8 | 52.2 | 77.1 | 25.0 | 51.4 | 53h |
| +MoPPS | 92.0 | 58.3 | 39.1 | 89.5 | 50.8 | 65.9 | 52.4 | 76.7 | 23.4 | 50.9 | 42h |
| +DS (Oracle) | 93.2 | 60.0 | 40.4 | 90.5 | 51.0 | 67.0 | 52.0 | 78.2 | 19.6 | 49.9 | 77h |
| +GPS (Ours) | 93.2 | 62.1 | 39.7 | 90.5 | 51.8 | **67.4** | 52.8 | 79.7 | 22.2 | **51.5** | 49h |

## 4.2. Main Results

### 4.2.1. RELIABLE DIFFICULTY PREDICTION

To evaluate the quality of difficulty prediction, we adopt *Spearman rank correlation coefficient* (Sedgwick, 2014) $\rho$, which measures the strength of a monotonic relationship between two sequences and is invariant to monotonic transformations, making it suitable for assessing relative difficulty ordering. Formally, it is defined as:

$$\rho = \frac{\mathrm{cov}\Big(\mathrm{rank}(\hat{\Gamma}^{\mathcal{B}}), \mathrm{rank}(\widetilde{\Gamma}^{\mathcal{B}})\Big)}{\sigma_{\mathrm{rank}(\hat{\Gamma}^{\mathcal{B}})} \cdot \sigma_{\mathrm{rank}(\widetilde{\Gamma}^{\mathcal{B}})}}, \qquad (16)$$

where $\hat{\Gamma}^{\mathcal{B}} = \{\hat{\gamma}^{\tau}\}_{\tau \in \mathcal{T}^{\mathcal{B}}}$ and $\widetilde{\Gamma}^{\mathcal{B}} = \{\widetilde{\gamma}^{\tau}\}_{\tau \in \mathcal{T}^{\mathcal{B}}}$ denote the predicted and empirical success rates for the batch $\mathcal{T}^{\mathcal{B}}$, and $\mathrm{rank}(\cdot)$ returns the rank ordering. Furthermore, we report the $p$-value under the null hypothesis testing that $\hat{\gamma}^{\tau}$ and $\widetilde{\gamma}^{\tau}$ are independent to assess statistical significance.

As shown in Fig. 2, GPS rapidly learns to predict prompt difficulty within only a few optimization steps, well before

completing a full epoch, achieving extremely low $p$-values. The correlation steadily improves as histories accumulate. This validates that the proposed generalizable PPM enables reliable difficulty prediction with cross-prompt generalization. Furthermore, Fig. 6 shows that GPS attains higher prediction quality than MoPPS and consistently higher effective sample ratio than Uniform, which can be attributed to our proposed comprehensive batch acquisition strategy.

### 4.2.2. EFFICIENT AND IMPROVED RLVR

To examine whether improved prompt selection translates into efficient RLVR training, we compare GPS with baselines across diverse scenarios and backbone models. Fig. 3 shows training curves with respect to training steps, while Tables 2 and 4 summarize the final performance. Overall, GPS enables both more efficient and effective RLVR training. Compared to Uniform, GPS consistently accelerates training, yielding a **1.4×–2.0×** speedup in terms of training steps. It also delivers average gains of 1.6–1.9 points on mathematical tasks and 4.1–5.7 points on logical tasks. These improvements come at minimal additional

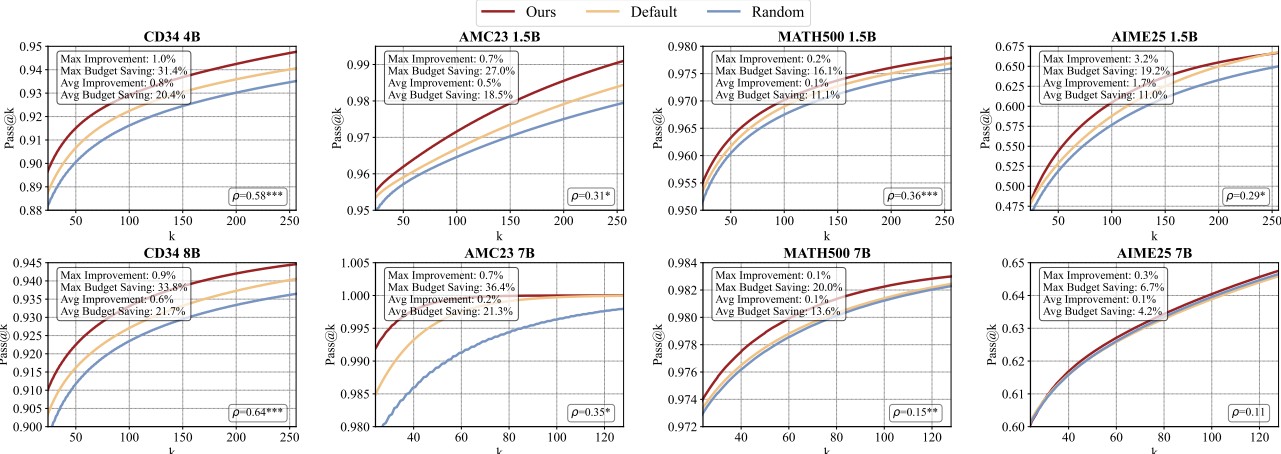

*Figure 4.* Test-time computation allocation results across benchmarks. pass@k versus the number of generated samples $k$ is shown, with insets reporting Spearman's rank correlation $\rho$ (* $p < 0.05$, ** $p < 0.01$, *** $p < 0.001$) and other statistical metrics.

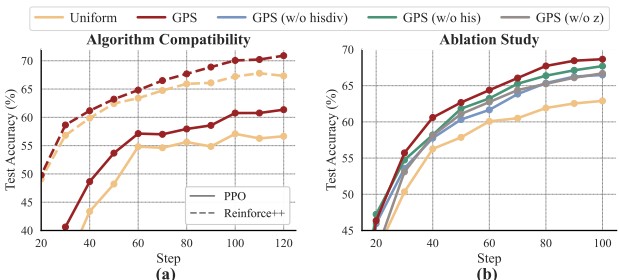

*Figure 5.* (a) Training curves on Countdown with PPO and Reinforce++. GPS is compatible with both algorithms and consistently outperforms Uniform. (b) Ablation study on Countdown, including removing history-anchored diversity (w/o hisdiv), removing inter-step exploration only (w/o his), and replacing the PPM with a deterministic PPM without latent variables (w/o $z$).

cost: the lightweight PPM introduces negligible overhead (Appendix E.1), and the modest increase in runtime is due to longer response generation (Fig. 12). Compared to DS, GPS achieves comparable performance while substantially reduced cost, requiring up to **69%** fewer rollouts (Fig. 7) and reducing training time by **28%–47%**. Benefiting from the design of generative PPM and unified prompt batch selection, GPS outperforms prediction-based baselines. Moreover, its strong performance on out-of-distribution benchmarks indicates improved general reasoning ability.

### 4.2.3. TEST-TIME GENERALIZATION AND EFFICIENCY

To investigate whether the learned generalizable PPM generalizes to test-time prompts, we conduct offline batch evaluations on logical and mathematical benchmarks. Fig. 4 shows that the PPM retains statistically significant correlation $\rho$ with empirical success rates on most unseen test benchmarks. Building on these predictions, we evaluate difficulty-guided computation allocation and compare against **Default** (fixed samples per prompt), and **Random** (the same alloca-

tion configurations with random difficulty predictions). As shown in Fig. 4, the proposed method yields up to a **3.2%** relative improvement over Default at a fixed budget, or reduces computation by up to **36.4%** with no loss in performance. Notably, this introduces no additional computational overhead. The results suggest that training-time difficulty prediction can be reused for test-time, enabling a coherent training–test pipeline. However, we observe that the predictive correlation naturally diminishes for semantically distant out-of-domain tasks, indicating that extending to broader domains may necessitate multi-domain co-training.

### 4.3. Additional Analysis

#### 4.3.1. ALGORITHM COMPATIBILITY

Although primarily evaluated with GRPO, GPS is compatible with other RLVR algorithms. We integrate it with two alternative algorithms, **PPO** (Schulman et al., 2017) and **Reinforce++** (Hu, 2025) on Countdown. As shown in Fig. 5(a), GPS consistently improves training efficiency and final performance over Uniform under both algorithms. Notably, GPS is compatible with PPO despite its single-response generation per prompt. This is enabled by the generalizable predictive model, in contrast to prior evaluation-based methods such as DS, which rely on multiple responses and are therefore inapplicable in this regime. These results position GPS as a broadly applicable solution for enhancing sample efficiency across diverse RLVR pipelines. GPS is also applicable to continuous-reward settings (see Appendix E.5).

#### 4.3.2. ABLATION STUDIES

We conduct ablation studies to evaluate the contributions of key components in GPS, including history-anchored diversity and the latent difficulty context in the generative PPM. As shown in Fig. 5(b), removing history-anchored diversity

(Ours w/o hisdiv) causes a substantial performance drop, while ablating only the inter-step exploration term (Ours w/o his) leads to a smaller decline. This indicates that both intra-batch dispersion and inter-step exploration play important roles in effective prompt batch selection. Replacing the generative PPM with a deterministic PPM that directly maps prompts to success rates, i.e., removing the latent difficulty context $z$ (Ours w/o z), also degrades performance, highlighting the benefit of modeling a shared latent difficulty context for cross-prompt generalization and policy evolution adaptation. Overall, these results confirm that each design component contributes meaningfully. Additional results are provided in Appendix E.6.

### 4.3.3. HYPERPARAMETERS

The effects of the diversity weight $\lambda$ and the candidate batch size are evaluated in Appendix E.9 and E.10.

## 5. Conclusion

This work presents Generalizable Predictive Prompt Selection to accelerate RL post-training of large reasoning models. The small generative PPM is effective enough in prompt difficulty estimate, allowing for cross-prompt generalization and adaptation to policy evolution. The acquisition of difficulty and batch-level diversity integration helps improve LLM and PPM training. Additionally, the learned PPM aids in test-time computation allocation.

**Discussion.** This work first verifies the feasibility of efficiently steering LLMs' RL post-training with small generative PPMs. Future work can be exploring more refined generative modeling for online prompt selection. In addition, this work has preliminarily revealed the opportunity of PPM-guided test-time computation allocation; more advanced designs may further enhance its effectiveness.

## Acknowledgment

Dr. Qi Wang acknowledges support from the National Natural Science Foundation of China (NSFC) under the grant number 62306326. This work was supported by the National Science and Technology Major Project 2025ZD1606303, Fundamental and Interdisciplinary Disciplines Breakthrough Plan of the Ministry of Education of China JYB2025XDXM503 and National Natural Science Foundation of China 62495092.

## Impact Statement

This work focuses on improving the computational efficiency of RL post-training for LLMs via online predictive prompt selection. By avoiding redundant rollouts, the proposed method helps lower training cost and resource consumption. The approach operates solely at the level of data selection and optimization strategy, without introducing new model capabilities or directly altering model outputs. Accordingly, it does not pose new societal or ethical risks beyond those inherent to LLMs, and any downstream impact depends on specific application contexts.

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

# Appendix Overview

This appendix provides supplementary discussions, theoretical analyses, and experimental details that support the main results. The appendix is organized as follows:

- **Appendix A (Related Works):** reviews related works on RL post-training of LLMs and online prompt selection for RLVR.

- **Appendix B (Additional Discussions):** offers clarifications on the contributions of the proposed method and further discusses the importance of history-anchored diversity, as well as the relationship between semantic representations and prompt difficulty.

- **Appendix C (Theoretical Proof):** presents detailed theoretical proof.

- **Appendix D (Implementation Details):** provides comprehensive implementation details, including tasks, models, training details, and the realization of the generalizable prompt predictive model, as well as its extension to test-time computation allocation.

- **Appendix E (Extended Experimental Results):** reports additional experimental results and analyses, including computational complexity analysis, extended evaluations, ablation studies, hyperparameter sensitivity analyses, training dynamics, and evaluations across different LLM families.

- **Appendix F (Data Examples):** presents representative data examples used in the experiments.

# A. Related Works

**RL Post-Training of LLMs.**  Reinforcement learning has become a central technique for aligning pretrained LLMs with desired behaviors and task objectives. Early successes are largely driven by Reinforcement Learning with Human Feedback (RLHF), which improves instruction following, safety, and alignment through preference-based optimization (Dong et al., 2024; Dai et al., 2023; Sun et al., 2023; Zheng et al., 2023). More recent progress highlights Reinforcement Learning with Verifiable Rewards (RLVR), where reward signals are automatically verified, leading to substantial gains in structured reasoning domains such as mathematics and programming (Jaech et al., 2024; Guo et al., 2025; Team et al., 2025; Chu et al., 2025; Pan et al., 2025; Luo et al., 2025a;b). From an algorithmic perspective, Proximal Policy Optimization (PPO) (Schulman et al., 2017) remains a standard choice, while Group Relative Policy Optimization (GRPO) (Shao et al., 2024) reduces computational overhead by eliminating the value network and estimating advantages through group-normalized rewards. Building on these foundations, a growing body of work focuses on stabilizing training, reducing bias, lowering variance, and improving sample efficiency (Yuan et al., 2025; Yue et al., 2025; Liu et al., 2025b; Yu et al., 2025; Kazemnejad et al., 2024; Hu, 2025; Zheng et al., 2025a; Qu et al., 2026). In parallel, large-scale empirical studies demonstrate the effectiveness of RL post-training across model sizes and application domains (Luo et al., 2025b; Dang & Ngo, 2025; Luo et al., 2025a; Zeng et al., 2025a; Meng et al., 2025; Xu et al., 2024), supported by increasingly mature systems for scalable RL training (Sheng et al., 2024; Zhu et al., 2025; Wang et al., 2025b).

**Online Prompt Selection for RLVR**  Beyond algorithmic improvements, data curation have emerged as critical factors in determining the efficiency of RL post-training. A line of offline approaches pre-select training prompts based on heuristics such as difficulty, diversity, or solution length (Ye et al., 2025; Li et al., 2025; Wen et al., 2025; Hu et al., 2025; Yang et al., 2024; Fatemi et al., 2025; Wang et al., 2025d; Zhang et al., 2022; Xiao et al., 2026). While effective in reducing training cost, these methods typically require additional preprocessing and remain static throughout training, making them fail to adapt to evolving LLM policy (Qu et al., 2025b; Gao et al., 2025). To address this limitation, recent work explores online prompt selection that adapt to the current policy. Some methods perform evaluation-based filtering by discarding uninformative prompts or prioritizing moderate difficulty (Yu et al., 2025; Liu et al., 2025a; Cui et al., 2025; Meng et al., 2025; Bae et al., 2025), often at the expense of additional LLM evaluations (Xu et al., 2026). An alternative direction is prediction-based prompt selection, which seeks to estimate prompt difficulty without explicit rollout-based evaluation (Zhang et al., 2025b; Zheng et al., 2025b; Gao et al., 2025; Qu et al., 2025b; Zeng et al., 2025b; Chen et al., 2025; Mao et al., 2026; Zou et al., 2025; Wang et al., 2025a; Qu et al., 2025c). A representative method is MoPPS (Qu et al., 2025b), which models each prompt independently using a prompt-specific Beta posterior to estimate its success rate based on historical observations. While this design avoids additional inference cost, it does not explicitly account for the continually evolving model state

during online optimization, nor does it allow information sharing across prompts. As a result, prompt-specific modeling may suffer from lagged adaptation and limited generalization, particularly in large prompt pools where many prompts are sparsely observed.

# B. Additional Discussions

## B.1. Contribution Clarification

Efficiently guiding data selection for LLM optimization is a fundamentally challenging problem with substantial practical significance (Wang et al., 2026). In RL post-training, evaluating candidate prompts using the target model itself is often prohibitively expensive, which makes evaluation-based strategies difficult to scale. As a result, designing mechanisms that can steer LLM optimization using limited auxiliary computation remains an open and practically important challenge.

Prior work has predominantly approached this problem through lightweight prompt-specific modeling or by leveraging large models to assess prompt difficulty. In contrast, this work represents an early attempt to use a shared and lightweight prompt predictive model to guide the optimization of a much larger LLM. This *small-steer-large* paradigm, in which a small predictive model captures transferable difficulty signals to drive LLM training, constitutes a key conceptual contribution and opens a promising design space for scalable optimization.

We emphasize that the specific choice of a conditional generative model for the prompt predictive model is not intended as a primary technical contribution. Rather, it serves as a concrete instantiation of the broader idea of shared, history-conditioned modeling. Our framework does not rely on the optimality of a particular generative architecture, and alternative formulations could be explored within the same paradigm. Empirically, we find that even a relatively simple generative predictive model is sufficient to extract useful signals for prompt selection, highlighting the potential of this direction rather than claiming architectural optimality.

## B.2. Discussion on the Importance of History-Anchored Diversity

The motivation for history-anchored diversity arises from the coupled nature of prompt selection, PPM learning, and LLM optimization. PPM predictions parameterize the acquisition criterion, which shapes the prompt distribution used for LLM optimization, while the optimization process in turn generates new feedback that is used to update the PPM. This creates an intrinsic feedback loop between sampling, prediction, and policy learning, analogous to a *chicken-and-egg* problem.

Without explicit diversity regularization, the acquisition criteria tends to over-exploit a narrow subset of prompts that are estimated to be most informative. While this can yield short-term gains, it may result in a progressively concentrated training distribution for the PPM, increasing the risk of overfitting and weakening its ability to generalize difficulty estimates to underexplored regions of the prompt space. As the predictor becomes increasingly biased toward frequently sampled prompts, such miscalibration can in turn reduce the diversity of training signals received by the LLM, potentially reinforcing sampling collapse and adversely affecting the performance of joint optimization.

History-anchored diversity explicitly counteracts this effect by encouraging coverage across both the current batch and historical selections. By maintaining a more balanced and diverse prompt distribution, it improves the effectiveness and robustness of the PPM. This, in turn, supports more effective LLM optimization through sustained exposure to diverse and appropriately challenging prompts. In Appendix E.6, the ablation results and the observed increase in the effective sample ratio provide empirical evidence for this role.

We note that history-anchored diversity is introduced as a general regularization principle rather than a specific instantiation, and alternative diversity measures or regularizers could be incorporated within the same framework.

## B.3. Discussion on Semantic Representations and Prompt Difficulty

Prompt difficulty is naturally determined jointly by the prompt itself and the evolving policy model. In our implementation, these two factors are explicitly considered: the former is represented by a fixed semantic embedding $\tau$, while the latter is captured by a time-evolving latent context $z_t$ that summarizes the optimization history. Formally, prompt difficulty $\gamma_t^\tau$ is modeled as $f(\tau, z_t)$.

To enable cross-prompt generalization, the framework implicitly relies on a weak structural assumption: under a fixed LLM policy, semantic representations provide a coarse organization of the prompt space in which difficulty is not entirely arbitrary.

Importantly, this does not require semantic similarity to be a strong or deterministic predictor of difficulty, nor does it imply that prompt embeddings alone can determine difficulty. Rather, semantic representations serve as a shared coordinate system that supports information transfer across prompts. Empirically, on DeepScaler, we conduct a simple baseline that estimates difficulty by weighting success rates using embedding similarity achieves a correlation of approximately $0.2$ under a fixed policy. While this level of correlation is insufficient for accurate difficulty prediction in isolation, it demonstrates that semantic structure provides non-trivial signals. At the same time, its weakness highlights the necessity of incorporating history-dependent difficulty context, motivating the use of the dynamically updated latent variable $z_t$.

Overall, semantic representations in our framework act as a stable but coarse anchor for cross-prompt generalization, while the latent context $z_t$ captures the dominant, time-varying factors induced by model optimization. Their combination enables effective prompt difficulty modeling. Finally, we emphasize that the specific choice of semantic embedding (e.g., WordLLaMA) is a practical instantiation rather than a conceptual requirement; the framework itself is agnostic to the particular embedding model used.

## C. Theoretical Proof

**Theorem 3.1** (Better Prediction with Shared History). *Let $\hat{\gamma}^{\tau,\mathrm{ind}} := \mathbb{E}[\gamma_t^\tau | H_{t-1}^\tau]$ be the optimal predictor based only on prompt-specific history, and $\hat{\gamma}^{\tau,\mathrm{shr}} := \mathbb{E}[\gamma_t^\tau | H_{t-1}]$ be the optimal predictor based on the full optimization history, where $H_{t-1}^\tau \subset H_{t-1}$. The prediction risk $\mathcal{R}(\hat{\gamma}) := \mathbb{E}[(\hat{\gamma} - \gamma_t^\tau)^2]$ admits the orthogonal decomposition*

$$\mathcal{R}(\hat{\gamma}^{\tau,\mathrm{shr}}) = \mathcal{R}(\hat{\gamma}^{\tau,\mathrm{ind}}) - \mathcal{C}(\tau), \tag{11}$$

*where $\mathcal{C}(\tau) = \mathbb{E}\left[\left(\hat{\gamma}^{\tau,\mathrm{shr}} - \hat{\gamma}^{\tau,\mathrm{ind}}\right)^2\right] \geq 0$ is the estimation gap. Moreover, $\mathcal{C}(\tau) > 0$ if and only if $H_{t-1}$ provides non-redundant predictive information about $\gamma_t^\tau$ beyond $H_{t-1}^\tau$.*

*Proof.* The proof follows a standard argument by interpreting conditional expectation as an orthogonal projection in the $L^2$ Hilbert space of random variables. We include it to formalize the role of shared history conditioning in our framework.

**Step 1: Orthogonality and MSE decomposition.** We decompose the independent estimation error as:

$$\gamma_t^\tau - \hat{\gamma}^{\tau,\mathrm{ind}} = (\gamma_t^\tau - \hat{\gamma}^{\tau,\mathrm{shr}}) + (\hat{\gamma}^{\tau,\mathrm{shr}} - \hat{\gamma}^{\tau,\mathrm{ind}}). \tag{17}$$

Let

$$\epsilon := \gamma_t^\tau - \hat{\gamma}^{\tau,\mathrm{shr}}, \qquad \delta := \hat{\gamma}^{\tau,\mathrm{shr}} - \hat{\gamma}^{\tau,\mathrm{ind}}.$$

Since $\hat{\gamma}^{\tau,\mathrm{shr}} = \mathbb{E}[\gamma_t^\tau \mid H_{t-1}]$, we have

$$\mathbb{E}[\epsilon \mid H_{t-1}] = \mathbb{E}[\gamma_t^\tau - \hat{\gamma}^{\tau,\mathrm{shr}} \mid H_{t-1}] = 0.$$

Moreover, $\delta$ is fully determined by $H_{t-1}$, so we can write

$$\mathbb{E}[\epsilon\,\delta] = \mathbb{E}\big[\mathbb{E}[\epsilon\,\delta \mid H_{t-1}]\big] = \mathbb{E}\big[\delta \cdot \mathbb{E}[\epsilon \mid H_{t-1}]\big] = 0. \tag{18}$$

The Pythagorean identity $\|\epsilon + \delta\|^2 = \|\epsilon\|^2 + \|\delta\|^2$ directly yields

$$\mathcal{R}(\hat{\gamma}^{\tau,\mathrm{ind}}) = \mathcal{R}(\hat{\gamma}^{\tau,\mathrm{shr}}) + \mathcal{C}(\tau), \tag{19}$$

where $\mathcal{C}(\tau) = \mathbb{E}[\delta^2] \geq 0$.

**Step 2: Risk reduction.** The term $\mathcal{C}(\tau) \geq 0$ represents the variance of the shared predictor unexplained by prompt-specific history. It is strictly positive if and only if $\hat{\gamma}^{\tau,\mathrm{shr}}$ cannot be determined solely from prompt-specific feedback, i.e., when the full optimization history provides additional predictive information about $\gamma_t^\tau$. □

*Remark* C.1 (Realization via Global Latent Context). The optimal shared estimator $\hat{\gamma}^{\tau,\mathrm{shr}}$ introduced above is generally intractable, as it requires conditioning on the full, high-dimensional optimization history $H_{t-1}$. Our framework therefore adopts a *practical surrogate* by encoding global difficulty context into a time-evolving latent variable $z_t$. Specifically,

the history-dependent prior $p_{\boldsymbol{\eta}}(\boldsymbol{z}_t \mid H_{t-1})$ is designed to capture coarse-grained properties of the optimization trajectory, while the shared decoder $p_{\boldsymbol{\psi}}(\gamma \mid \tau, \boldsymbol{z}_t)$ enables experience sharing across prompts. This construction is motivated by the theoretical advantage of shared conditioning, without assuming that the latent representation fully characterizes the optimization history. In practice, even a coarse contextual signal can be sufficient to provide a reasonably stable relative ordering for prompt selection, which is the primary objective of our framework. Empirically, we observe a consistent correlation between predicted and observed prompt difficulty, supporting this behavior.

# D. Implementation Details

## D.1. Tasks

### D.1.1. MATHEMATICS REASONING

**Training Dataset.** Following Luo et al. (2025b); Gao et al. (2025), we train models on the DeepScaler dataset (Luo et al., 2025b), which consists of 40,315 competition-level mathematics problems. We use the public version hosted at `https://huggingface.co/datasets/agentica-org/DeepScaleR-Preview-Dataset`.

**Evaluation Benchmarks.** We evaluate mathematical reasoning performance on a suite of benchmarks, including AIME24, AMC23, MATH500 (Lightman et al., 2023), Minerva Math (Lewkowycz et al., 2022), and OlympiadBench (He et al., 2024), using the datasets hosted at `https://huggingface.co/datasets/math-ai`. Training curves report the average performance across all math benchmarks. To further assess generalization beyond the training distribution, we additionally evaluate on three general reasoning benchmarks: MMLU-Pro (Wang et al., 2024b), ARC-c (Clark et al., 2018), and GPQA-diamond (GPQA) (Rein et al., 2024), using datasets provided by LUFFY (Yan et al., 2025).

**Reward Function.** Following the default configuration in verl (Sheng et al., 2024), we adopt a binary reward scheme that assigns a reward of 1 to correct responses and 0 otherwise.

### D.1.2. LOGICAL REASONING

**Training Dataset.** We consider the Countdown Number Game, a symbolic reasoning task that requires combining given numbers via basic arithmetic operations to reach a target value. For training, we use a 20,000-problem subset of the Countdown-34 dataset from `https://huggingface.co/datasets/Jiayi-Pan/Countdown-Tasks-3to4`, where each problem provides either three or four source numbers.

**Evaluation Benchmarks.** Evaluation is conducted on two held-out benchmarks: a 512-problem split from Countdown-34 (CD-34), and a 512-problem subset from Countdown-4 (CD-4), a more challenging variant available at `https://huggingface.co/datasets/Jiayi-Pan/Countdown-Tasks-4`. Compared to CD-34, CD-4 consistently provides four source numbers per instance, substantially enlarging the search space and increasing problem difficulty. Training curves are reported on the average performance on CD-34 and CD-4.

**Reward Function.** Following Pan et al. (2025), we incorporate a format-aware reward function:

$$r = \begin{cases} 1 & \text{if the response is correct,} \\ 0.1 & \text{if the response is incorrect but properly formatted,} \\ 0 & \text{otherwise.} \end{cases} \tag{20}$$

## D.2. Models

We evaluate five LLMs spanning different architectures and parameter scales. All models are obtained from their official Hugging Face repositories and used without modification:

- DeepSeek-R1-Distill-Qwen-1.5B: `https://huggingface.co/deepseek-ai/DeepSeek-R1-Distill-Qwen-1.5B`;

- DeepSeek-R1-Distill-Qwen-7B: `https://huggingface.co/deepseek-ai/DeepSeek-R1-Distill-Qwen-7B`;

- Qwen3-4B-Base: `https://huggingface.co/Qwen/Qwen3-4B-Base`;

- Qwen3-8B-Base: `https://huggingface.co/Qwen/Qwen3-8B-Base`;

- Llama-3.2-3B-Instruct: `https://huggingface.co/meta-llama/Llama-3.2-3B-Instruct`.

### D.3. Training Details

We adopt GRPO (Shao et al., 2024) as the default RLVR algorithm, implemented within the verl framework (Sheng et al., 2024). At each training step, $k = 8$ responses are sampled per prompt to estimate advantages, using temperature 1.0 and `top_p` $= 1.0$. Evaluation is performed using `pass@1`, computed from multiple independent generations per prompt, where the number of generations varies across benchmarks. Following Luo et al. (2025b); Qu et al. (2025b), evaluation generations use temperature 0.6 and `top_p` $= 0.95$. We disable the KL penalty by setting $\beta = 0$, consistent with Yu et al. (2025).

The training batch size $\mathcal{B}$ is set to 256 for both DeepScaler and Countdown, with mini-batch sizes of 128 and 64, respectively. The maximum response length is 8192 tokens for DeepScaler and 1024 tokens for Countdown. Entropy regularization is applied with a coefficient of 0.001 for Countdown and disabled for DeepScaler. Optimization is performed using Adam (Kingma & Ba, 2014) with learning rates of $1e{-}6$ for Countdown and $4e{-}6$ for DeepScaler, following Qu et al. (2025b); Gao et al. (2025), with $\beta = (0.9, 0.999)$, and weight decay 0.01. We apply the Clip-Higher strategy from DAPO (Yu et al., 2025), which decouples clipping ranges with $\epsilon_{\text{low}} = 0.2$ and $\epsilon_{\text{high}} = 0.28$. For Countdown, we apply a post-rollout advantage normalization to stabilize gradient magnitudes under low effective sample ratios. For each benchmark, training is conducted under a fixed computational budget; in practice, this corresponds to roughly one pass over the dataset, by which point performance has largely stabilized. All experiments are conducted on 8 NVIDIA H20 GPUs.

Regarding hyperparameters, we set $\lambda = 1$ for Countdown and $\lambda = 0.5$ for DeepScaler. The candidate batch consists of the entire prompt pool. Regarding the diversity implementation in Eq. (13), we use Euclidean distance in the embedding space as a simple and computationally efficient distance measure.

### D.4. Implementation of the Generalizable Prompt Predictive Model

We implement the prompt predictive model using a variational formulation optimized via the ELBO in Eq. (8), jointly learning the encoder, decoder, and history-conditioned prior.

**Input Representation.** Prompt embeddings are extracted using the WORDLLAMA toolkit (Miller, 2024), selected for its computational efficiency and stable semantic representations.

**Encoder and Prior.** The variational encoder $q_{\boldsymbol{\phi}}(\boldsymbol{z}_t \mid H_t)$ and the history-conditioned prior $p_{\boldsymbol{\eta}}(\boldsymbol{z}_t \mid H_{t-1})$ are implemented using lightweight Transformer encoders. They operate over short sequences that aggregate prompt embeddings and their associated success rate feedback. For computational efficiency, the prior conditions on the most recent training batch, while the encoder conditions on the current batch. This design leverages the recursive structure of the latent formulation: information from recent history is progressively compressed into the latent state through posterior-to-prior regularization, while longer-term effects are reflected in the evolving parameters of the predictive model as a result of training on preceding data. Consequently, conditioning on recent observations suffices to capture short-term non-stationarity in the optimization trajectory without explicitly re-encoding the full history at each step. The Transformer outputs are pooled to parameterize the mean and variance of a Gaussian latent context $\boldsymbol{z}_t$.

**Decoder.** The shared decoder $p_{\boldsymbol{\psi}}(H_t \mid \boldsymbol{z}_t)$ is parameterized as a multilayer perceptron (MLP), which maps the latent context $\boldsymbol{z}_t$ together with the prompt embedding $\tau_t$ to a predicted difficulty $\hat{\gamma}_t^{\tau_t}$. This shared parameterization enables experience transfer across prompts while keeping the predictive model lightweight.

Overall, this design remains consistent across all experimental settings and LLM backbones, enabling an efficient and stable implementation. The model introduces negligible computational overhead, with only **20**M parameters, less than **1**% of most LLMs. The detailed computational complexity analysis is deferred to Appendix E.1.

**Discussion.** The design choices of the predictive model, including the specific variational formulation, Transformer-based encoder, and lightweight MLP decoder, are not claimed to be optimal. Rather, the primary goal of this work is to validate the feasibility and effectiveness of global, history-conditioned difficulty modeling for prompt difficulty prediction under

realistic computational constraints. Accordingly, we intentionally adopt simple and stable architectures without extensive architecture search or scaling studies. Exploring alternative model classes, varied predictive model capacities, or more sophisticated temporal modeling strategies may further improve predictive accuracy, but such investigations are orthogonal to the central contributions of this paper and are left for future work.

## D.5. Extension to Test-Time Computation Allocation

After LLMs post-training, reasoning performance can be further improved by scaling test-time computation (Snell et al., 2024; Fu et al., 2025). A common instantiation is *best-of-N* sampling (Cobbe et al., 2021; Lightman et al., 2023), which generates $N$ independent responses per prompt and selects the best one. Under a binary correctness reward, performance is measured by the $\mathrm{pass@k}$ metric:

$$\mathrm{pass@k}(\tau) = 1 - (1 - \gamma^\tau)^k. \tag{21}$$

**Necessity of Computation Allocation.**   Typical test-time scaling methods allocate a fixed number of samples to each prompt, i.e., $k(\tau) \equiv k$ for all $\tau \in \mathcal{T}$. Under a global computation budget, this is suboptimal since the marginal benefit of additional samples varies across prompts (Damani et al., 2024; Wang et al., 2025c). Specifically, the marginal gain of one extra sample to prompt $\tau$ is

$$\Delta_k(\tau) = \mathrm{pass@(k+1)}(\tau) - \mathrm{pass@k}(\tau) = (1 - \gamma^\tau)^k \gamma^\tau. \tag{22}$$

As $k$ increases, easy prompts quickly saturate, and extremely hard ones with $\gamma^\tau \to 0$ yield negligible gains. Challenging yet solvable prompts retain the largest marginal gains, which motivates difficulty-guided computation allocation.

**Computation Allocation with Predicted Difficulty.**   Accurately estimating $\gamma^\tau$ at test time is costly and hinders parallelism. Prior methods propose pre-training predictors (Damani et al., 2024; Snell et al., 2024) or sequential pre-sampling (Raman et al., 2025) to estimate the prompt difficulty. We reuse the PPM learned during RLVR, providing a new scheme for test-time allocation. Independent modeling methods such as MoPPS do not generalize to unseen benchmarks and are therefore unsuitable in this setting.

Specifically, we formulate test-time computation allocation as a difficulty quantile-based function. Given an evaluation set $\mathcal{T}$ and a global budget $\sum_{\tau \in \mathcal{T}} k(\tau) = k \cdot |\mathcal{T}|$, each prompt is assigned a minimum number of samples $k_{\min}$, and the remaining computation is distributed according to

$$k(\tau) = k_{\min} + \alpha \cdot w(q(\hat{\gamma}^\tau), k), \tag{23}$$

where $q(\hat{\gamma}^\tau) \in [0, 1]$ denotes the empirical quantile of $\hat{\gamma}^\tau$ within $\{\hat{\gamma}^{\tau'}\}_{\tau' \in \mathcal{T}}$, $w(\cdot)$ is an allocation function defined over quantiles, and $\alpha$ is determined by the budget constraint. This formulation depends only on relative ordering and is therefore robust to miscalibration on unseen benchmarks.

Inspired by Damani et al. (2024), we adopt a simple window function in this work,

$$w(q(\hat{\gamma}^\tau), k) = \mathbb{1}[q_1(k) < q(\hat{\gamma}^\tau) < q_2(k)], \tag{24}$$

which provides a coarse approximation to the marginal utility structure of $\mathrm{pass@k}$: prompts that are either too easy or unsolvable receive little computation, while challenging yet solvable prompts are prioritized.

**Scope and Implementation Notes.**   We emphasize that the detailed computation allocation strategy is *not* a technical contribution of this work, nor is it intended to be optimal. Our goal here is solely to demonstrate that a generalizable PPM, once learned during RLVR, can be naturally reused at test time to guide computation allocation. The specific functional form of $w(\cdot)$ and the choice of hyperparameters are deliberately kept simple and are adopted as a concrete instantiation rather than a carefully optimized design, and primarily serve to instantiate this idea in practice. The empirical results should therefore be interpreted as evidence of the applicability of generalizable PPMs to test-time computation allocation, rather than as an exhaustive study of optimal allocation strategies. In practice, we set $k_{\min} = 0.1 \cdot k$ across all scenarios. The quantile thresholds are selected via lightweight calibration under low-computation regimes: we evaluate $\mathrm{pass@k}$ over a small range of $k$ values and choose thresholds that yield stable improvements. While not optimal, this lightweight calibration is to demonstrate the feasibility of difficulty-guided allocation in practice. Across benchmarks and model sizes, the resulting thresholds $(q_1, q_2)$ typically focus on difficult prompts (e.g., $(0, 0.5)$ for CD34 and $(0, 0.6)$ for MATH500). In settings with substantial distribution shift, an interesting direction for future work is to treat the learned PPM as a pretrained initialization and adapt it via lightweight finetuning before test-time allocation.

# E. Extended Experimental Results

## E.1. Computational Complexity Analysis

Table 3 summarizes the per-step runtime of different operations for Countdown and DeepScaler under varied LLM scales. For reference, we report the per-step average runtime of LLM training and rollout generation under uniform prompt sampling. Evaluation-based methods such as DS incur additional overhead by performing rollout-based evaluation over enlarged candidate sets; specifically, the DS sampling cost is computed as the rollout multiplicative factor relative to the standard batch size, multiplied by the average rollout time. In contrast, the proposed GPS performs prompt selection using a lightweight predictive model and does not require any additional LLM inference. As a result, the extra cost introduced by GPS remains negligible ($< 1\%$) compared to the overall cost of LLM training and rollout generation.

Additionally, to assess scalability under large candidate pools, we evaluate GPS on a synthetic prompt set containing $1M$ prompts, with a candidate batch size of the same scale. In this setting, the combined cost of GPS sampling and PPM updates remains $< 16s$, indicating that the overhead grows sublinearly and remains negligible relative to LLM training. Notably, the current implementation runs the PPM on a single GPU and further acceleration is straightforward via data or model parallelism in large-scale settings. Moreover, the candidate batch size can be chosen substantially smaller to further reduce computational cost without affecting performance, as shown in Appendix E.10.

*Table 3.* Per-step computational cost of different operations across tasks and LLM scales. All measurements are taken during RL post-training on $8\times$H20 GPUs with batch size 256.

| Component | Countdown | | DeepScaler | |
|---|---|---|---|---|
| | **4B** | **8B** | **1.5B** | **7B** |
| LLM Training | $\sim$72 s | $\sim$116 s | $\sim$203 s | $\sim$610 s |
| LLM Rollout | $\sim$32 s | $\sim$39 s | $\sim$157 s | $\sim$290 s |
| DS Sample Cost | $\sim$3$\times$32 s | $\sim$3$\times$39 s | $\sim$3$\times$157 s | $\sim$3.6$\times$290 s |
| GPS (Sample + PPM Update) | $\sim$1 s | $\sim$1 s | $\sim$1.6 s | $\sim$1.6 s |

## E.2. Additional Evaluation Results

Table 4 report the final evaluation results on Countdown.

*Table 4.* Evaluation results on Countdown. '+' denotes finetuning with the corresponding method. 'Avg.' reports the average accuracy, and 'Runtime' indicates total training time. **Bold** and underlined values indicate the best and second-best results, respectively.

| Method | CD4 Avg@16 | CD34 Avg@16 | Avg.↑ | Runtime↓ |
|---|---|---|---|---|
| **Qwen3-4B** | 1.3 | 3.5 | 2.4 | - |
| **+Uniform** | 51.1 | 73.8 | 62.5 | 2.9h |
| **+PCL** | 51.0 | 72.8 | 61.9 | 3.2h |
| **+GRESO** | 53.8 | 73.8 | 63.8 | 4.9h |
| **+MoPPS** | 52.9 | 73.9 | 63.4 | 2.8h |
| **+DS** | 56.1 | 76.3 | 66.2 | 4.9h |
| **+GPS (Ours)** | 57.2 | 76.0 | **66.6** | 3.4h |
| **Qwen3-8B** | 2.1 | 3.9 | 3.0 | - |
| **+Uniform** | 52.5 | 73.3 | 62.9 | 4.3h |
| **+PCL** | 53.5 | 74.9 | 64.2 | 5.1h |
| **+GRESO** | 54.1 | 75.1 | 64.6 | 6.2h |
| **+MoPPS** | 55.5 | 76.0 | 65.7 | 4.3h |
| **+DS** | 58.7 | 78.2 | 68.5 | 6.9h |
| **+GPS (Ours)** | 59.4 | 77.9 | **68.6** | 5.0h |

## E.3. Quality of Difficulty Prediction and Prompt Selection

To further demonstrate that the generalizable PPM yields more accurate predictions than prompt-specific difficulty modeling, we compare Spearman's rank correlation with the representative prompt-specific method MoPPS. Notably, to isolate

prediction quality from sampling-induced distributional effects, all correlations are computed under uniform prompt sampling. As shown in the top row of Fig. 6, GPS consistently achieves higher correlation than MoPPS. In contrast, MoPPS struggles to provide meaningful predictions under large-scale training, due to the sparsity of prompt-specific observations. These results indicate that the proposed generalizable PPM enables more accurate and stable difficulty estimation.

To assess prompt selection effectiveness, we further report the *Effective Sample Ratio* (ESR) and compare against Uniform and the prompt-specific baseline MoPPS. ESR is defined as the fraction of prompts with non-zero reward variance:

$$\text{ESR}(\mathcal{T}^{\mathcal{B}}) = \frac{1}{\mathcal{B}} \sum_{\tau \in \mathcal{T}^{\mathcal{B}}} \mathbf{1}\Big[\text{Var}(\{r_j^{\tau}\}_{j=1}^{k}) > 0\Big], \tag{25}$$

where $\mathbf{1}[\cdot]$ is the indicator function. A higher ESR indicates more informative prompts in the batch.

As shown in the bottom row of Fig. 6, GPS maintains a consistently higher ESR throughout training, which is attributed to the reliable difficulty prediction and unified batch selection strategy. This indicates that GPS selects more informative prompt batches, which in turn contributes to its superior performance.

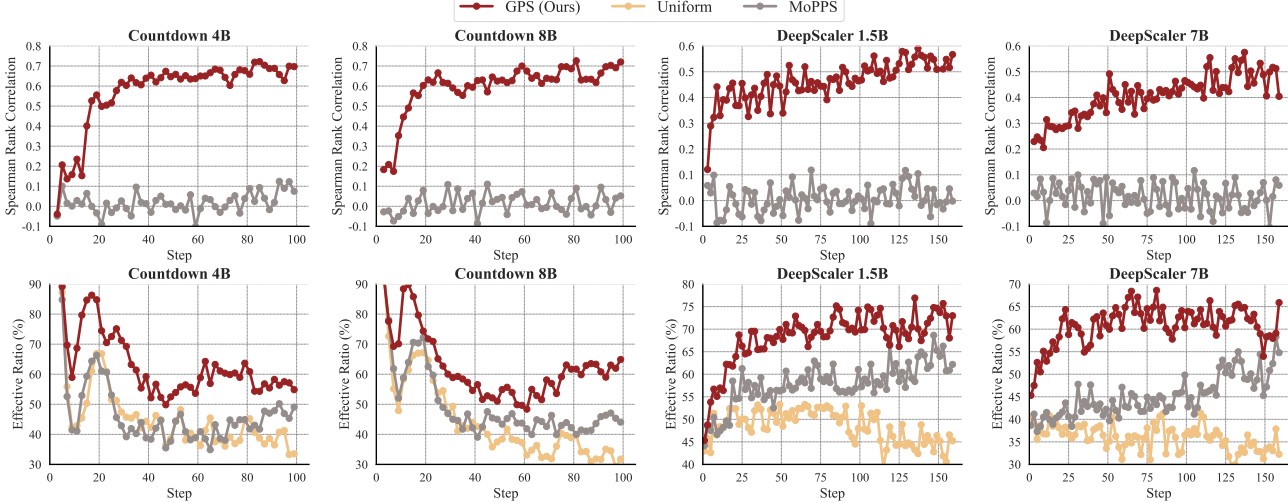

*Figure 6.* Difficulty prediction quality and effective sample ratio during training. The top row shows Spearman's rank correlation between predicted difficulty and empirical success rate. The bottom row reports the effective sample ratio achieved by different online prompt selection methods.

### E.4. Performance versus Rollout Number

To explicitly illustrate the rollout efficiency of GPS relative to evaluation-based methods such as DS, we plot training performance against the number of generated rollouts. As shown in Fig. 7, prediction-based methods such as GPS introduce no additional rollouts beyond those required for training, whereas evaluation-based method DS incur substantially higher rollout costs due to the evaluation of larger candidate prompt sets. GPS achieves comparable performance while requiring only **31%–34%** of the rollouts used by DS. These results highlight the rollout efficiency enabled by prompt difficulty prediction.

### E.5. Applicability to Continuous Process Rewards

Most prior works on online prompt selection focus on tasks with binary rewards, which is also the primary setting studied in this work. Nevertheless, the proposed prompt predictive model does not rely on the binary reward assumption and is applicable to more general reward functions. However, an open challenge lies in defining appropriate acquisition criteria for selecting informative prompts when rewards are continuous and process-based. In contrast to binary rewards, where prompts with intermediate difficulty $\gamma \approx 0.5$ are most informative, there is no canonical target value for continuous rewards.

Inspired by prior studies (Xu et al., 2025; Razin et al., 2025), we adopt a simple yet effective hypothesis: prompts that exhibit higher reward variance across rollouts provide more informative learning signals. Notably, this criterion naturally subsumes

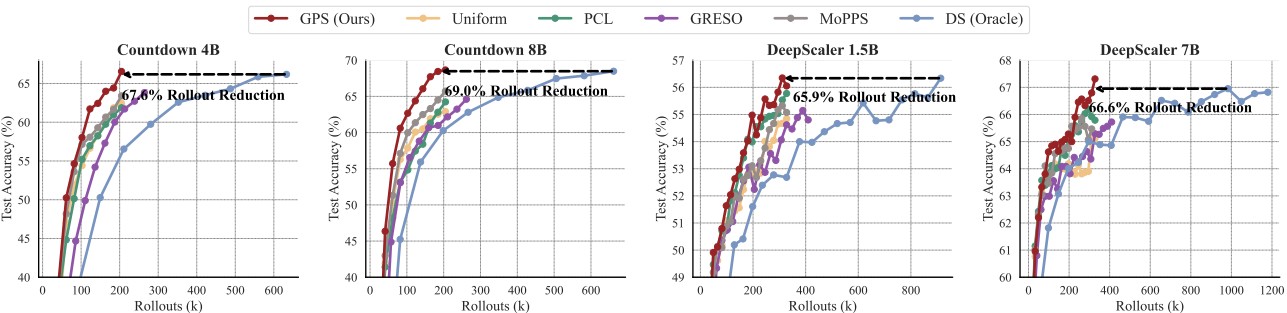

*Figure 7.* Training curves of GPS and baseline methods across different scenarios and backbone models, plotted against the number of generated rollouts to reflect computational overhead.

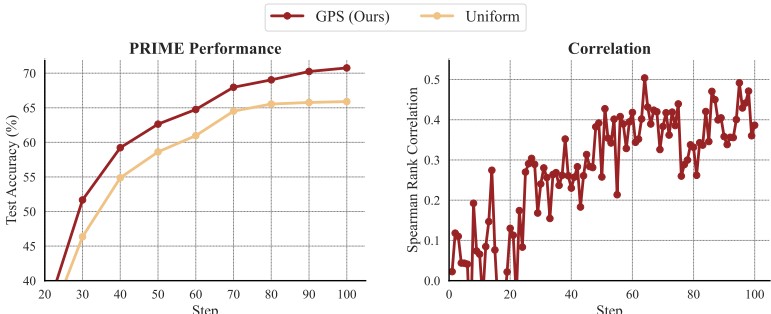

*Figure 8.* Evaluation on Countdown with PRIME using continuous process rewards. GPS achieves consistent improvements over uniform prompt selection and demonstrates reliable reward-variance prediction, indicating its applicability beyond binary reward settings.

the binary-reward case, where reward variance is maximized at $\gamma = 0.5$. Consequently, we replace difficulty-based utility $u(\hat{\gamma}^\tau)$ in Eq. (12) with reward variance $\mathrm{Var}(\boldsymbol{r}^\tau)$, prioritizing prompts with greater variance in continuous process rewards. Correspondingly, the prediction target of the PPM is changed from the success rate $\gamma^\tau$ to the reward variance $\mathrm{Var}(\boldsymbol{r}^\tau)$.

To empirically assess the applicability of GPS beyond binary feedback, we conduct a preliminary study by integrating our method with PRIME (Cui et al., 2025), which provides continuous process rewards. As shown in Fig. 8, GPS consistently outperforms uniform prompt selection under continuous process rewards and maintains reliable prediction of reward variance. These results provide initial evidence that the proposed framework extends beyond binary reward settings and can accommodate more general forms of feedback.

### E.6. Additional Ablation Study and the Role of History-Anchored Diversity

In addition to the ablation study on Countdown presented in the main paper, we further conduct the same set of ablations on the DeepScaler. As shown in Fig. 9, the relative performance trends on DeepScaler closely mirror those observed on Countdown. In particular, removing history-anchored diversity (GPS w/o hisdiv) results in a pronounced performance degradation, while ablating the latent difficulty context in the generative predictive model (GPS w/o z) also consistently harms performance. These results confirm that both components contribute robustly across datasets.

To more directly examine the role of history-anchored diversity, we further evaluate its impact on the effective sample ratio. As shown in Fig. 10, introducing history-anchored diversity consistently increases ESR. This observation supports the explanation given in the main text: by explicitly encouraging diversity, history-anchored diversity mitigates overfitting of the PPM, leading to more reliable difficulty estimation and more effective prompt selection.

### E.7. Evaluation with Different LLM families

Beyond the Qwen and DeepSeek series, we evaluate GPS and baseline methods on the Countdown using Llama-3.2-3B-Instruct, an instruction-tuned model from the LLaMA family that differs substantially in architectural design and training paradigm. As shown in Fig. 11, GPS continues to achieve reliable difficulty prediction and consistently better performance.

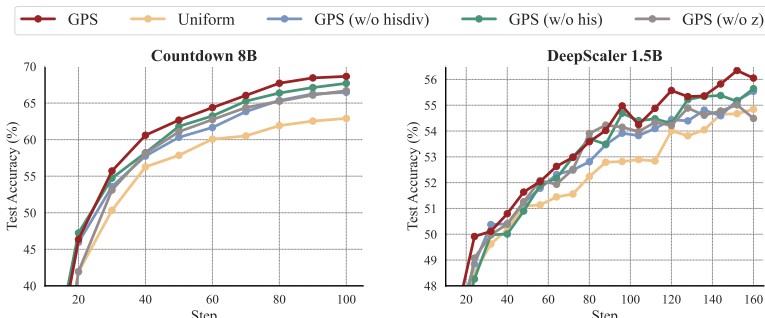

*Figure 9.* Ablation study of key components in GPS on Countdown and DeepScaler, including the generative predictive model and history-anchored diversity.

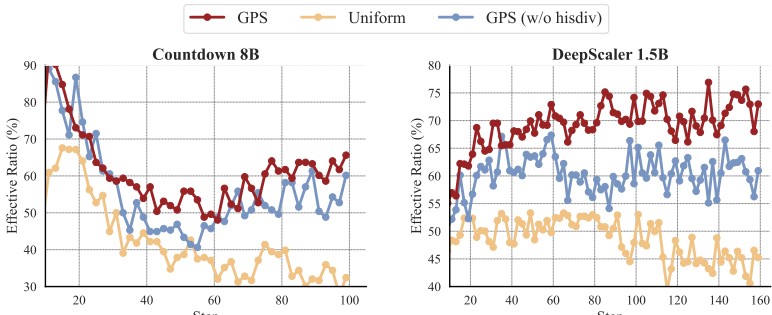

*Figure 10.* Effect of history-anchored diversity on the effective sample ratio during training.

These results indicate that the proposed method is not specialized to a particular LLM series, but has the potential of generalizing across heterogeneous model families.

### E.8. Training Dynamics

Fig. 12 illustrates the training dynamics of response length, entropy, and training reward under different prompt selection strategies.

**Response Length.** Regarding response length, the proposed method exhibits a trend highly similar to DS, while both consistently produce longer responses than uniform sampling. This suggests that sampling prompts of intermediate difficulty tends to elicit longer reasoning chains. This effect also explains the longer wall-clock training time of our method compared to uniform sampling.

**Entropy.** For entropy, no single monotonic pattern emerges across all settings. In the DeepScaler experiments, both DS and GPS lead to increased entropy during training, indicating enhanced exploration. Notably, our method consistently induces a stronger entropy increase, suggesting a more diverse exploration behavior, which we leave for future investigation.

**Training Rewards.** Finally, training rewards align well with expectations. Compared to uniform sampling, GPS maintains the average reward close to $0.5$ throughout training, reflecting a sustained focus on informative prompts. This behavior empirically validates the effectiveness of the proposed difficulty estimation and prompt selection strategy.

### E.9. Sensitivity to Diversity Weight $\lambda$

We study the effect of diversity weight $\lambda$ in GPS, which controls the difficulty-diversity trade-off. As shown in Fig. 13a, very small values of $\lambda$ underemphasize diversity, resulting in redundant prompt selection and limited coverage of the prompt space. In contrast, excessively large $\lambda$ over-prioritizes diversity at the expense of utility, leading to degraded training performance. Importantly, GPS exhibits relatively stable performance over a broad intermediate range of $\lambda$, indicating robustness to the choice of this hyperparameter.

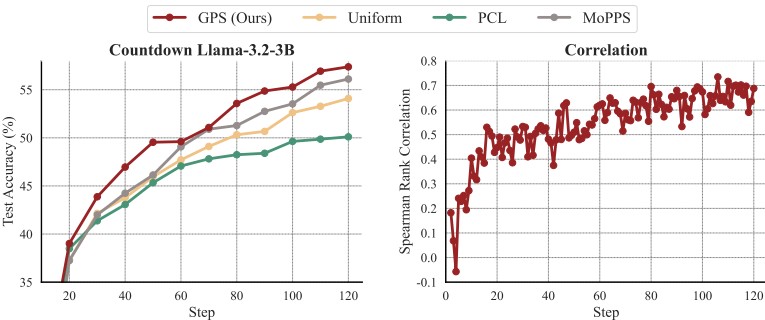

*Figure 11.* Evaluation on Countdown with Llama-3.2-3B-Instruct, demonstrating the effectiveness of the proposed method across different LLM families.

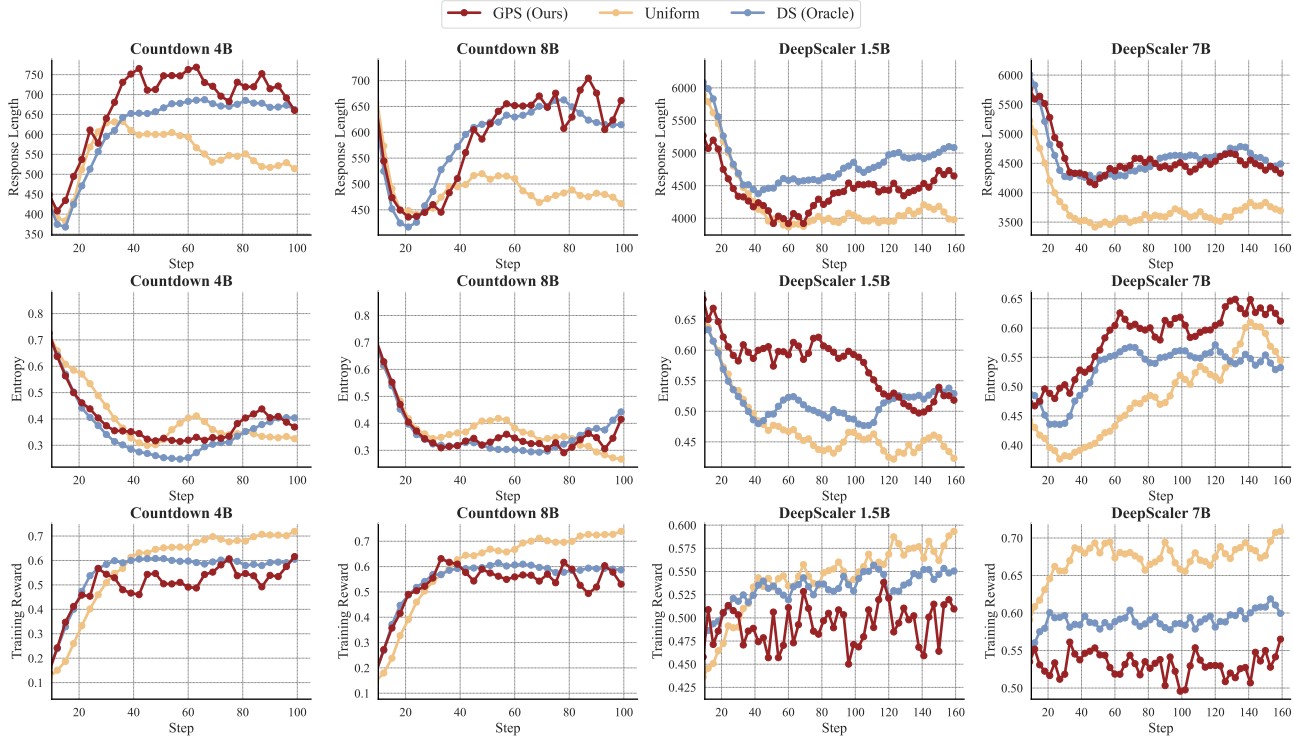

*Figure 12.* Training dynamics, showing response length, entropy, and training reward throughout training.

### E.10. Effect of Candidate Batch Size

We further investigate the effect of the candidate ratio, defined as the ratio between the candidate batch size and the actual batch size used for training. By default, we use the full prompt pool as the candidate set, which incurs negligible overhead due to the lightweight PPM. As shown in Fig. 13b, increasing the candidate batch size tends to improve performance, as a larger pool provides more flexibility for prompt batch selection. The gains gradually saturate once the candidate size exceeds a moderate threshold, suggesting diminishing returns beyond this point. This trend is consistent with prior observations (Qu et al., 2025b).

### E.11. Extension to Mixed-Domain Training

To demonstrate the robustness and versatility of our method beyond single-domain scenarios, we conduct an additional experiment on mixed-domain training, incorporating both mathematical reasoning and code generation tasks. Specifically, we train the Qwen3-1.7B-Base model on a composite dataset comprising DeepScaler (Luo et al., 2025b) for mathematics and Prime Code (Cui et al., 2025) for programming. The results in Table 5 show that GPS consistently outperforms Uniform in both domains, indicating that our method is not limited to narrow tasks, but generalizes effectively to broader, mixed-domain

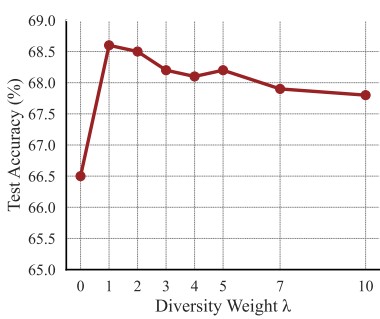
*(a)* Effect of diversity weight $\lambda$.

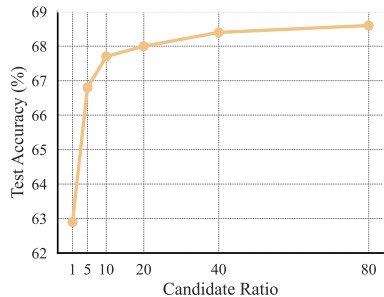
*(b)* Effect of candidate batch size.

*Figure 13.* Hyperparameter sensitivity analysis on Countdown 8B.

*Table 5.* Evaluation on mathematics and coding benchmarks.

| Method | Math | | | | | | Code | | |
|---|---|---|---|---|---|---|---|---|---|
| | MATH500 Avg@1 | Olympiad. Avg@1 | Minerva. Avg@4 | AMC23 Avg@32 | AIME24 Avg@32 | Avg.↑ | CodeContests Avg@4 | Codeforces Avg@4 | Avg.↑ |
| Uniform | 65.2 | 24.7 | 25.0 | 36.5 | 6.1 | 31.5 | 23.8 | 24.8 | 24.3 |
| GPS (Ours) | 65.6 | 28.6 | 24.0 | 40.6 | 11.2 | **34.0** | 26.9 | 26.0 | **26.5** |

settings including programming tasks.

### E.12. Sensitivity Analysis across Embedding Models

GPS is orthogonal to the choice of embedding model: the specific embedding is an implementation detail, rather than a prerequisite for our core contribution. To assess the sensitivity of our framework, we evaluate its performance using multiple widely adopted embedding models on Countdown.

As shown in Table 6, while the absolute performance varies slightly depending on the embedding used, GPS consistently and significantly outperforms the uniform sampling baseline across all choices. Specifically, although WordLlama achieves the highest accuracy, alternative models such as all-MiniLM-L6-v2 and bge-large-en-v1.5 (Xiao et al., 2023) still yield substantial improvements over the baseline. These results indicate that the empirical gains of GPS are robust to the specific embedding choice and fundamentally stem from the proposed generalizable prompt selection framework.

*Table 6.* Sensitivity analysis across different embedding models on Countdown with Qwen3-8B model.

| Task | Uniform | GPS with Various Embeddings | | |
|---|---|---|---|---|
| | | WordLlama | all-MiniLM-L6-v2 | bge-large-en-v1.5 |
| Countdown 8B | 62.9 | **68.6** | 67.9 | 67.7 |

### E.13. Applicability beyond Reasoning: Evaluation on RLHF

To demonstrate the applicability of GPS beyond reasoning tasks with binary rewards, we evaluate it in an RLHF setting using the HH-RLHF (Bai et al., 2022) dataset and Qwen3-1.7B-Base. To accommodate continuous rewards, we adapt our PPM to predict the average continuous reward and employ a quantile-threshold sampling strategy. As shown in Table 7, GPS maintains a strong Spearman correlation with empirical rewards. By dynamically prioritizing informative prompts, GPS achieves a $1.24\times$ training speedup and improves final performance compared to the uniform sampling baseline. While this heuristic proves effective, establishing a theoretically optimal sampling criterion for continuous rewards remains a promising direction for future exploration.

*Table 7.* GPS reliably predicts continuous rewards and significantly accelerates training compared to uniform on RLHF.

| Method | Spearman Correlation | Speedup | Performance |
|--------|----------------------|---------|-------------|
| **Uniform** | – | $1.00\times$ | 17.4 |
| **GPS** (Ours) | $0.7\sim0.8$ | **$1.24\times$** | **18.1** |

### E.14. Discussion on Future Directions

While the current study focuses primarily on online RL-based post-training, the proposed sampling perspective may extend more broadly to offline preference optimization or off-policy training pipelines (Mao et al., 2024a; Dong et al., 2024; Mao et al., 2024b; Qu et al., 2023; Shao et al., 2023), where efficient allocation of optimization effort across data remains an important challenge. Another important future direction is evaluating the resulting models in more complex downstream environments, particularly agentic and interactive scenarios (Jin et al., 2025a; Qu et al., 2024; Wang et al., 2024a; Qu et al., 2025a). Beyond standard reasoning benchmarks, such evaluations could provide additional insights into how adaptive sampling strategies influence long-horizon planning, tool use, environment interaction, and practical deployment performance.

## F. Data Examples

For the DeepScaler dataset and mathematics benchmarks, prompts are constructed by appending a chain-of-thought instruction (Wei et al., 2022) together with a formatting constraint: "`Let's think step by step and output the final answer within \boxed{}`". For general reasoning benchmarks, we follow the evaluation protocol of Yan et al. (2025) and adopt PRIME's prompt template. For the Countdown task, we use the prompt format introduced in Pan et al. (2025).

---

**DeepScaler & Mathematics Benchmarks example**

**Prompt:**
The operation $\otimes$ is defined for all nonzero numbers by $a \otimes b = \frac{a^2}{b}$. Determine $[(1 \otimes 2) \otimes 3] - [1 \otimes (2 \otimes 3)]$. Let's think step by step and output the final answer within \boxed{}.
**Ground-Truth Answer:**
$-\frac{2}{3}$

---

**General Reasoning Benchmarks example**

**Prompt:**
The following are multiple choice questions (with answers) about $. Think step by step and then finish your answer with "\boxed{X}" where X is the correct letter choice.
Question:
Typical advertising regulatory bodies suggest, for example that adverts must not: encourage ________, cause unnecessary _______ or ____, and must not cause ______ offence.
Options:
A. Unsafe practices, Wants, Jealousy, Serious
B. Unsafe practices, Distress, Joy, Trivial
C. Unsafe practices, Distress, Fear, Serious
D. Safe practices, Wants, Jealousy, Trivial
E. Unsafe practices, Wants, Fear, Trivial
F. Safe practices, Wants, Fear, Serious
G. Safe practices, Distress, Fear, Trivial
H. Safe practices, Distress, Jealousy, Serious
I. Safe practices, Fear, Jealousy, Trivial
**Ground-Truth Answer:**
C

---

**Countdown example**

**Prompt:**
A conversation between User and Assistant. The user asks a question, and the Assistant solves it. The assistant first thinks about the reasoning process in the mind and then provides the user with the answer.
User: Using the numbers [2, 54, 17], create an equation that equals 35. You can use basic arithmetic operations (+, -, *, /) and each number can only be used once. Show your work in <think> < /think> tags. And return the final answer in <answer> < /answer> tags, for example <answer> $(1 + 2)/3$ </answer>.
Assistant: Let me solve this step by step.
<think>

