# OpenReview forum: "Small Generalizable Prompt Predictive Models Can Steer Efficient RL Post-Training of Large Reasoning Models"
_ICML.cc/2026/Conference — ICML 2026 regular_

### Official Review · Reviewer_X2BT · 2026-03-13

**Soundness:** 3
**Presentation:** 3
**Significance:** 2
**Originality:** 2
**Overall Recommendation:** 4
**Confidence:** 4

**Summary:**

This paper investigates optimizing the prompt selection for efficient RL post-training. The authors introduces a method named Predictive Prompt Selection (GPS), which is trained on shared optimization history to estimate prompt difficulty. Though the approach of difficulty estimation exists in literature, its Bayesian inference approach is interesting. In terms of empirical performance, GPS reliably predicts prompt difficulty, and accelerates the training by up to 2x through prompt selection.

**Compliance With Llm Reviewing Policy:**

Affirmed.

**Key Questions For Authors:**

1. Can you provide more insight or empirical evidence on how GPS avoids overfitting to the optimization history? How does the estimation accuracy evolve during the training?
2. I am interested to see examples of how GPS handle prompts that are semantically very different from those seen during training.
3. Is the method applicable to other domains beyond reasoning tasks? Will the difficulty estimation be easier or more difficult? I wonder whether the magnitude of the speedup is similar to that in the reasoning task.

**Limitations:**

yes

**Strengths And Weaknesses:**

Strengths

1. The proposed approach is principled, with established Bayesian inference framework and batch-level considerations. Theoretical justifications are provided.
2. The method leverages shared optimization history, enabling accurate difficulty estimation even for rarely seen / unseen prompts.

Weaknesses

1. Only logical reasoning and math tasks are evaluated in this paper.
2. There lack principled justifications that the predicted difficulty corresponds to useful learning signals for the RL objective

---

> ### Author Rebuttal · Authors · 2026-03-31
>
> *We thank the reviewer X2BT for the careful reading and thoughtful comments, which help us improve the quality of our work. Below we answer the questions in detail.*
>
> ---
> > **Q1. Extension to Diverse Tasks**
>
> Thanks for the valuable comment.
> We conduct **mixed-domain training (math + code)** by training Qwen3-1.7B-Base on DeepScaler [1] (math) + Prime Code[2].
>
> |Method|||Math||||||Code||
> |-|:-:|:-:|:-:|:-:|:-:|:-:|:-:|:-:|:-:|:-:|
> ||MATH500|Olympiad.|Minerva.|AMC23|AIME24|**Avg.**||CodeContests|Codeforces|**Avg.**|
> ||Avg@1|Avg@1|Avg@4|Avg@32|Avg@32|||Avg@4|Avg@4||
> |**Uniform**|65.2|24.7|25.0|36.5|6.1|31.5||23.8|24.8|24.3|
> |**Ours**|65.6|28.6|24.0|40.6|11.2|**34.0**||26.9|26.0|**26.5**|
>
> The results show that GPS is **not limited to narrow tasks**, but generalizes to **broader, mixed-domain settings including programming**.
> We will add these in Appendix E.
>
> > **Q2. Connection between Predicted Difficulty and Learning Signal**
>
> Thanks for the thoughtful comment.
> We clarify that using success rate as a difficulty proxy and targeting 0.5 is **not our core contribution** but a concrete instantiation for binary-reward RLVR. This is well-supported:
> - **Intuition:** Intermediate-difficulty prompts naturally induce an easy-to-hard curriculum.
> - **Theory:** 0.5 success rate maximizes reward variance, yielding stronger learning signals and optimization potential[3,4].
> - **Evidence:** Prior works find intermediate-difficulty prompts most effective[5,6].
>
> Importantly, our primary contribution is the **Generalizable Predictive Prompt Selection paradigm**, which is flexible and **accommodates alternative criteria**.
> For instance, beyond binary rewards, we extended GPS to track and maximize reward variance (Appendix E.5).
> We will add the discussion in Sec. 3.3 of the revision.
>
> > **Q3. Insight and Empirical Evidence on Avoiding Overfitting**
>
> Thanks for the insightful question.
> - **Insight:** GPS tracks prompt difficulty via streaming variational inference. Rather than directly fitting history, GPS compresses observations into a shared global latent variable $z_t$, representing the LLM's current capability. The adaptive $z_t$ acts as a compact information bottleneck, forcing the predictor to generalize through a shared mapping across prompts rather than memorizing history. The difficulty–diversity acquisition further encourages prompt space coverage, mitigating overfitting.
> - **Evidence:** Fig. 1 shows that the Spearman correlation (representing estimation accuracy) **steadily improves during training**, indicating effective adaptation without overfitting. Fig. 10 shows diversity-aware acquisition enhances selection and prevents PPM overfitting.
>
> We will add the discussion in Appendix B of the revision.
>
> > **Q4. Generalization to Semantically Distant Prompts**
>
> Thanks for the insightful question, which helps clarify GPS's generalization scope.
> - **In-domain:** Fig. 4 shows that the learned PPM generalizes well to unseen prompts within the same domain, maintaining a significant correlation.
> - **Out-of-domain (OOD):** For semantically distant OOD tasks (Countdown, ARC-c) with a PPM trained on DeepScaleR, we observe that predicted difficulty correlates poorly with empirical success.
>
> This is expected: GPS relies on semantic representations as a coarse organization of the prompt space.
> When semantic structure differs, predictive power drops.
> Extending GPS to broader domains may require multi-domain co-training or more universal representations.
> We will add the discussion in Sec. 4.2.3.
>
> > **Q5. Applicability beyond Reasoning**
>
> Thanks for the thoughtful suggestion.
> Beyond reasoning, we test GPS on **RLHF** with the HH-RLHF dataset and Qwen3-1.7B-Base.
> Because RLHF uses continuous rewards, we predict the average reward and adopt a quantile-threshold sampling.
>
> |RLHF|Spearman corr|Speedup|Performance|
> |-|:-:|:-:|:-:|
> |**Uniform**|-|-|17.4|
> |**Ours**|$0.7 \sim 0.8$ |**1.24x**|**18.1**|
>
> Results indicate that GPS can reliably predict difficulty and accelerate training in non-reasoning domains.
> Optimal sampling criterion for continuous rewards remains an open question, suggesting future exploration.
> We will add this in Appendix E of the revised manuscript.
>
>
> **References:**\
> [1] DeepScaleR: Effective RL Scaling of Reasoning Models via Iterative Context Lengthening\
> [2] Process reinforcement through implicit rewards\
> [3] Improving Data Efficiency for LLM Reinforcement Fine-tuning Through Difficulty-targeted Online Data Selection and Rollout Replay\
> [4] CurES: From Gradient Analysis to Efficient Curriculum Learning for Reasoning LLMs \
> [5] Can prompt difficulty be online predicted for accelerating rl finetuning of reasoning models?\
> [6] Prompt curriculum learning for efficient llm post-training
>
> ---
> *Thanks again for your time and constructive comments. We hope our responses have clarified the key points. The code will be released to support further research. We would be happy to address any further questions.*

---

> > ### Author Rebuttal · Reviewer_X2BT · 2026-04-03
> >
> > I thank the authors for the responses, I will keep my positive score.

---

> > > ### Author Response · Authors · 2026-04-03
> > >
> > > We sincerely appreciate **Reviewer X2BT** and the **Area Chair** for their time and efforts throughout the review and rebuttal process. The constructive feedback and insightful questions are invaluable to improving the manuscript. We will incorporate these suggestions and the discussed results into the updated version.
> > > Thanks.

---

### Official Review · Reviewer_GtF6 · 2026-03-13

**Soundness:** 3
**Presentation:** 3
**Significance:** 2
**Originality:** 2
**Overall Recommendation:** 4
**Confidence:** 2

**Summary:**

This paper studies online prompt selection for RL post-training of reasoning language models. It proposes GPS, a small generalizable prompt predictive model that uses shared optimization history to estimate prompt difficulty, combined with a diversity-aware batch selection strategy that prioritizes informative intermediate-difficulty prompts. Experiments on mathematical and logical reasoning tasks show that GPS improves RL training efficiency and final performance over prior selection methods, while the same predictor also generalizes to unseen test prompts for difficulty-aware compute allocation at inference time.

**Compliance With Llm Reviewing Policy:**

Affirmed.

**Final Justification:**

The paper is basically sound and empirically solid. After the rebuttal, most of my concerns were addressed. I also agree with the points mentioned by other reviews, e.g. the task scope is narrow. Overall, I would like to maintain my positive score for this submission.

**Key Questions For Authors:**

See above.

**Limitations:**

yes

**Strengths And Weaknesses:**

## Strengths
* Tackles an important and timely problem: improving sample efficiency in RL post-training for reasoning LLMs.
* Broad and fairly solid empirical evaluation across training domains, model sizes, baselines, RL algorithms, and test-time allocation settings.
* GPS improves both training efficiency and final performance, and the same learned predictor is reused effectively at test time, providing practical contributions:
* Useful ablations and prediction-quality analysis that support the value of both the latent predictive model and the diversity-aware selection design.

## Weaknesses
* The method has several interacting components, so the overall contribution feels somewhat system-like and less conceptually minimal.  While the empirical results are good, the method itself is somewhat complex, with several interacting ingredients. Because of this, it is somewhat harder to isolate which conceptual component is truly essential beyond what is shown in the ablations. The ablations are helpful, but the full method still feels more like a well-engineered system than a sharply minimal principle.
* The paper’s notion of “good prompts are those with difficulty near 0.5” is intuitive, but it is still a somewhat stylized assumption. Its theoretical justification is limited beyond the binary-reward setting.
* The paper is stronger empirically than theoretically. The latent generative formulation is reasonable, but the paper does not provide a deeper analysis of why this history-based predictor should be expected to generalize as well as it does, nor when the batch acquisition rule should be optimal. This does not invalidate the method, but it does make the conceptual foundation somewhat less strong than the empirical section.

---

> ### Author Rebuttal · Authors · 2026-03-31
>
> *We sincerely thank the reviewer GtF6 for the careful reading and constructive feedback. We appreciate the recognition of the importance and the practical contributions. Below we answer the questions in detail.*
>
> ---
>
> > **Q1. Clarification on Conceptual Contribution and Design Principle**
>
> Thanks for the thoughtful comment.
> We clarify that GPS follows the **small-steer-large paradigm (Appendix B.1), using a lightweight, generalizable prompt predictive model to guide a larger LLM's optimization**.
> **The components are derived from this principle**, namely a shared, history-based difficulty modeling mechanism and a principled batch acquisition strategy to improve sampling quality:
> - **Generalizable Prompt Predictive Model** is the foundation and derived via streaming variational inference to predict current difficulty from global observation history (Sec. 3.2), enabling uncertainty-aware estimation and cross-prompt generalization.
> - **Difficulty-Diversity Batch Selection** bridges difficulty prediction and LLM optimization, and arises from optimizing batch-level sampling for informative and diverse prompts.
>   - Difficulty prioritizes informative samples.
>   - Diversity arises from the coupling between prompt selection, PPM learning, and LLM optimization, preventing redundancy and improving prompt space coverage for better PPM training and LLM optimization (Appendix B.2)
>
> Empirically, ablations in Sec. 4.3.2 show the effectiveness and necessity.
> Importantly, the design can be viewed as a concrete instantiation of our Generalizable Predictive Prompt Selection paradigm, which defines the core conceptual contribution.
> We expect this principle to open a broader design space for more minimal or efficient future implementations
>
> We will revise the paper to emphasize the contribution.
>
> > **Q2. Clarification on Intermediate-Difficulty Prioritization.**
>
> Thanks for the valuable comment.
> We clarify that targeting prompts with difficulty near 0.5 is not a central assumption of GPS, but **one concrete instantiation, motivated by prior works [1–5]**. GPS is flexible and can accommodate alternative sampling criteria.
>
> For the common binary-reward RLVR, this choice is well-supported:
> - **Intuition:** Selecting intermediate-difficulty prompts naturally induce an easy-to-hard curriculum.
> - **Theory:** 0.5 success rate maximizes reward variance, yielding stronger learning signals and optimization potential [1–3].
> - **Evidence:** Prior works consistently find intermediate-difficulty prompts most effective [4,5].
>
> **GPS can adapt to other criteria**. For example, beyond binary rewards, we explore a natural extension based on maximizing reward variance in Appendix E.5. Extending theoretical justification to general rewards remains an interesting future direction.
>
> We will add this discussion in Sec. 3.3 of the revised manuscript.
>
> > **Q3. Analysis of Designs: Generalization and Batch Acquisition**
>
> Thanks for the insightful comment. We will strengthen the conceptual discussion in the revision as follows:
>
> - **Generalization of the history-based predictor:**
> The history-based predictor generalizes via a structured decomposition of prompt difficulty: $\gamma_t^\tau = f(\tau, z_t)$ (Appendix B.3). $\tau$ (prompt embeddings) provides a shared semantic space across prompts, while $z_t$ (global latent variable) captures the evolving state of the LLM.
> This encourages the PPM to learn a mapping from semantic space to difficulty that adapts with the global model state, enabling principled generalization to unseen prompts.
>
> - **Rationale for the Difficulty-Diversity acquisition rule:**
> The acquisition rule is not claimed to be globally optimal, but serves as a practical surrogate for the coupled system of prompt selection, PPM learning, and LLM optimization (Appendix B.2).
> Optimizing only for difficulty risks over-exploit a small subset of prompts, causing PPM overfitting and redundant training signals for the LLM.
> By jointly considering difficulty and diversity, the acquisition rule maintains informative selection while improving coverage of the prompt space. This promotes stable co-optimization of PPM and LLM, as supported by our ablation studies (Appendix E.6).
>
>
> **References:**\
> [1] Improving Data Efficiency for LLM Reinforcement Fine-tuning Through Difficulty-targeted Online Data Selection and Rollout Replay\
> [2] Self-Evolving Curriculum for LLM Reasoning \
> [3] CurES: From Gradient Analysis to Efficient Curriculum Learning for Reasoning LLMs \
> [4] Can prompt difficulty be online predicted for accelerating rl finetuning of reasoning models?\
> [5] Prompt curriculum learning for efficient llm post-training
>
> ---
> *Thanks again for your time and constructive comments. We hope our responses have clarified the key points. The code will be released to support further research. We would be happy to address any further questions.*

---

> > ### Author Rebuttal · Reviewer_GtF6 · 2026-04-03
> >
> > I appreciate the authors for their detailed explanations, and they helped me better understand the contribution of this work. I would like to maintain my positive score.

---

> > > ### Author Response · Authors · 2026-04-03
> > >
> > > We sincerely thank **Reviewer GtF6** and the **Area Chair** for their time and efforts throughout the review and rebuttal process. The constructive comments and insightful questions are very helpful for improving the manuscript. We will incorporate these suggestions and discussions into the updated version. Thank you.

---

### Official Review · Reviewer_hevB · 2026-03-13

**Soundness:** 3
**Presentation:** 3
**Significance:** 2
**Originality:** 3
**Overall Recommendation:** 4
**Confidence:** 3

**Summary:**

This paper proposes GPS, a framework for efficient online prompt selection in RLVR, motivated by the fact that the RL training bottleneck is the rollout cost of generating and verifying long CoT rollouts. The authors address this by using a lightweight generative prompt predictive model that predicts prompt difficulty while leveraging shared optimization history via a global latent variable. The paper also introduces an unified batch acquisition objective that prioritizes intermediate-difficulty examples while explicitly encouraging diversity both within a batch and across steps. Empirically, GPS achieves substantial efficiency gains (up to ~2x speedup and large rollout reductions) while maintaining competitive or oracle level accuracy on the downstream tasks, and additionally demonstrates that the PPM generalizes to unseen tasks at test time for compute allocation.

**Compliance With Llm Reviewing Policy:**

Affirmed.

**Key Questions For Authors:**

The learning dynamic of models of different size and family can vary a lot, would this affect the learnability of prompt difficulty for PPM?

**Limitations:**

yes

**Strengths And Weaknesses:**

### Strength:
* The paper is well motivated, addressing an important bottleneck in modern RLVR training, the generation and verification cost of rollouts.
* The method is simple and intuitive, pluggable to existing RL pipeline, and the results show clear compute savings without sacrificing performance.
* The idea of prompt predictive model trained via ELBO optimization is novel and elegant, and the small steering large paradigm is appealing


### Weakness:
* Many recent RL training pipelines (e.g., ProRL) often conduct mixed-domain training. It is unclear whether PPM can robustly adapt and predict prompt difficulty when prompts come from different domains.
* The PPM requires prompt embeddings as input for prompt difficulty prediction. It is unclear whether the quality of these embeddings would be a bottleneck (e.g., common text embedding models might not yield good enough embeddings for code in rarer languages)



[1] Liu, Mingjie et al. “ProRL: Prolonged Reinforcement Learning Expands Reasoning Boundaries in Large Language Models.” ArXiv abs/2505.24864 (2025): n. pag.

---

> ### Author Rebuttal · Authors · 2026-03-31
>
> *We sincerely thank the reviewer hevB for the careful reading, constructive feedback, and positive assessment of our work. We are pleased that you find the proposed method simple, effective, and well-motivated. Below we answer the questions in detail.*
>
> ---
>
> > **Q1. Extension to Mixed-Domain Training**
>
> Thanks for this important suggestion. We evaluate GPS under **mixed-domain training (math + code)** by training Qwen3-1.7B-Base on DeepScaler [1] (math) + Prime Code [2].
>
> **Difficulty Prediction**: GPS remains effective across domains: the **Spearman correlation** between predicted difficulty and empirical success rate is approximately **0.5-0.6 (p-values significant)**. The **effective sample ratio** increases from **40% to 55%** compared to uniform sampling, indicating that GPS can still reliably prioritize informative prompts under mixed-domain settings.
>
> **Performance Gains**: This improvement in sampling quality translates into consistent gains in final performance:
>
> |Method|||Math||||||Code||
> |-|:-:|:-:|:-:|:-:|:-:|:-:|:-:|:-:|:-:|:-:|
> ||MATH500|Olympiad|Minerva|AMC23|AIME24|**Avg.**||CodeContests|Codeforces|**Avg.**|
> ||Avg@1|Avg@1|Avg@4|Avg@32|Avg@32|||Avg@4|Avg@4||
> |**Uniform**|65.2|24.7|25.0|36.5|6.1|31.5||23.8|24.8|24.3|
> |**Ours**|65.6|28.6|24.0|40.6|11.2|**34.0**||26.9|26.0|**26.5**|
>
> These results demonstrate that GPS generalizes effectively to **broader, mixed-domain settings**.
>
> We will include these additional results in Appendix E of the revision to strengthen the empirical validation of scalability.
>
>
> > **Q2. Discussion on Prompt Embeddings**
>
> We thank the reviewer for the insightful question.
>
> **(a) Sensitivity analysis across embedding models**
> As noted in Line 779-780, GPS is **orthogonal to the choice of embedding model**: the specific embedding is an implementation detail, not part of our core contribution.
> We **evaluate multiple embedding models**:
>
> |Countdown 8B|WordLlama|all-MiniLM-L6-v2|bge-large-en-v1.5|
> |-|-|-|-|
> |62.9 (Uniform)|68.6|67.9|67.7|
>
> While performance varies slightly, GPS **consistently outperforms baselines across embedding choices**, demonstrating robustness to the embedding model.
> These results indicate that the gains of GPS do not depend on a specific embedding choice, but stem from the proposed generalizable prompt selection framework.
>
> **(b) Robustness to code in rare languages**
> Our understanding is that PPM operates on **prompt-level semantic embeddings**, where prompts are typically natural language descriptions even for code tasks. Therefore, as discussed in Appendix B.3, embeddings are only required to **capture a coarse organization of the prompt space**, not detailed code syntax.
> In the **extreme case where embeddings are uninformative**, PPM naturally degenerates to prompt-specific baselines rather than failing, providing additional robustness.
>
> Overall, these results and discussions suggest that while proper embeddings may improve performance, they are **not a bottleneck** for GPS.
>
> We will incorporate this discussion in Appendix B.3 of the revised manuscript.
>
> > **Q3. Effect of varying learning dynamics across model sizes and families**
>
> We thank the reviewer for this insightful question.
>
> **Empirically**, we have evaluated GPS across multiple model sizes (1.5B to 8B) and families (Qwen, DeepSeek, LLaMA). GPS remains consistently effective and stable, indicating **robustness to varying learning dynamics**.
>
> **Conceptually**, estimating prompt difficulty can be viewed as tracking a temporal process, i.e., the evolving success rate.
> Different model families or sizes may exhibit different dynamics, but this does **not fundamentally affect the learnability** of PPM.
> Instead, it relies on the underlying structure of these dynamics, e.g., temporal continuity and prompt-level correlations.
> GPS is designed to track the evolution of difficulty using a PPM derived via streaming variational inference, without being coupled to any specific model architecture.
> Accordingly, as long as the learning process exhibits basic temporal smoothness and shared cross-prompt structure, GPS is expected to remain effective at tracking its evolution.
>
> We will include this discussion in Appendix E.7 of the revised manuscript.
>
>
> **References:**
> [1] DeepScaleR: Effective RL Scaling of Reasoning Models via Iterative Context Lengthening\
> [2] Process reinforcement through implicit rewards
>
> ---
> *Thanks again for your time and constructive comments. We hope our responses have clarified the key points. The code will be released to support further research. We would be happy to address any further questions.*

---

> > ### Author Rebuttal · Reviewer_hevB · 2026-04-04
> >
> > Thanks authors for the thorough response. Most of my concerns have been addressed, and I will maintain my score.

---

> > > ### Author Response · Authors · 2026-04-04
> > >
> > > We sincerely appreciate **Reviewer hevB** and the **Area Chair** for their time and efforts throughout the review and rebuttal process. The constructive feedback and insightful questions are invaluable to improving the manuscript. We will incorporate these suggestions and the discussed results into the updated version. Thanks.

---

### Official Review · Reviewer_vWeT · 2026-03-24

**Soundness:** 3
**Presentation:** 3
**Significance:** 4
**Originality:** 4
**Overall Recommendation:** 5
**Confidence:** 3

**Summary:**

The paper introduced Generalizable Predictive Prompt Selection (GPS) by training a lightweight generative model to predict prompt difficulty, which is then used to select intermediate difficulty prompts for training batches for better training and show more efficient compute allocation during test time. When GPS was used during training, we see significant improvement in training efficiency.

**Compliance With Llm Reviewing Policy:**

Affirmed.

**Key Questions For Authors:**

NA, see weaknesses.

**Limitations:**

Yes

**Strengths And Weaknesses:**

Strengths:
- The paper is well-written and well motivated, and address a real problem in LLM RL.
- Very solid experiments and comprehensive baseline which led to strong results that supports paper's claim on 1) PPM learns to estimate prompt difficulty in a generalizable way, 2) PPM supported RL post-training is more efficient, and 3) supports the test-time computation allocation.

Weaknesses:
- The problems covered in the paper is narrow - count down, math and STEM QA and the response length is rather small (8192). A more diverse set of tasks and bigger compute budget are needed to confirm the scalability of such results.

---

> ### Author Rebuttal · Authors · 2026-03-31
>
> *We sincerely thank the reviewer vWeT for the careful evaluation and thoughtful feedback. We are pleased that you find our work meaningful and appreciate the positive assessment. Below we provide detailed responses to your questions.*
>
> ---
> **Q1. The problems covered in the paper is narrow - count down, math and STEM QA and the response length is rather small (8192). A more diverse set of tasks and bigger compute budget are needed to confirm the scalability of such results.**
>
> We thank the reviewer for this important suggestion. We have conducted additional experiments:
>
> **(1) Scaling to larger compute budgets.**
> We train DeepSeek-R1-Distill-1.5B using the JustRL recipe [1] on the DAPO-17k dataset, with **substantially increased response length** (16k for training and 32k for evaluation on AIME24) and a **longer training horizon** (1500 steps).
>
> |Method|AIME24 Avg@32|Speedup|
> |-|:-:|:-:|
> |Uniform|45.8|-|
> |Ours|**48.9**|**1.5×**|
>
> Our method consistently outperforms uniform sampling, while achieving an approximate **1.5× training speedup**, demonstrating that the proposed approach remains effective and efficient as compute scales.
>
> ---
>
> **(2) Extension to diverse tasks.**
> We conduct **mixed-domain training (math + code)** by training Qwen3-1.7B-Base on a combination of DeepScaler [2] (math) and Prime Code [3] datasets.
>
> |Method|||Math||||||Code||
> |-|:-:|:-:|:-:|:-:|:-:|:-:|:-:|:-:|:-:|:-:|
> ||MATH500|Olympiad.|Minerva.|AMC23|AIME24|**Avg.**||CodeContests|Codeforces|**Avg.**|
> ||Avg@1|Avg@1|Avg@4|Avg@32|Avg@32|||Avg@4|Avg@4||
> |**Uniform**|65.2|24.7|25.0|36.5|6.1|31.5||23.8|24.8|24.3|
> |**Ours**|65.6|28.6|24.0|40.6|11.2|**34.0**||26.9|26.0|**26.5**|
>
> The results show that our method is **not limited to narrow tasks**, but generalizes effectively to **broader, mixed-domain settings including programming tasks**.
>
>
> We will include these additional results in Appendix E of the revision to strengthen the empirical validation of scalability.
>
> **References:**
> [1] JustRL: Scaling a 1.5B LLM with a Simple RL Recipe\
> [2] DeepScaleR: Effective RL Scaling of Reasoning Models via Iterative Context Lengthening\
> [3] Process reinforcement through implicit rewards
>
> ---
> *Thanks again for your time and constructive comments. We hope our responses have clarified the key points. The code will be released to support further research. We would be happy to address any further questions.*

---

> > ### Author Rebuttal · Reviewer_vWeT · 2026-04-04
> >
> > The authors have addressed my questions and comments! I will adjust my score.

---

> > > ### Author Response · Authors · 2026-04-04
> > >
> > > We sincerely appreciate **Reviewer vWeT** and the **Area Chair** for their time and efforts throughout the review and rebuttal process. We especially appreciate the recognition of our work, as well as the constructive feedback and insightful questions, which are invaluable for improving the manuscript. We will incorporate these suggestions and the discussed results into the revised version. Thanks.

---

### Decision · Program_Chairs · 2026-04-30

**Decision:**

Accept (regular)

**Comment:**

The reviewers were unanimously positive about this paper, and this positive assessment remained after the rebuttal. The paper addresses an important and timely problem in RL post-training for reasoning LLMs: reducing rollout and verification cost through better prompt selection. Reviewers found the method technically sound, well motivated, and practically relevant. In particular, they highlighted the lightweight and generalizable prompt predictive model, the diversity-aware batch selection strategy, and the strong empirical evidence that GPS improves training efficiency while also supporting test-time compute allocation.

The main concerns were about the scope and conceptual grounding of the work: the initial evaluation focused mostly on reasoning-style tasks, the full method has several interacting components, and the justification for the intermediate-difficulty criterion is more empirically than theoretically. Reviewers also asked about robustness across domains, embedding choices, and model families. In the rebuttal, the authors provided substantial additional evidence and clarifications, including larger-compute experiments, mixed-domain results, discussion of robustness across embeddings and model families, and clarification of the design rationale and generalization mechanism. Reviewers indicated that these responses adequately addressed their concerns and maintained their positive recommendations.

Overall, the paper presents a useful and nontrivial contribution on improving the efficiency of RL post-training for reasoning models. While some theoretical questions and broader-scope validation remain open, the current submission is technically solid, well presented, and supported by convincing empirical results. I therefore support acceptance.